# Mitochondrial CaMKII causes adverse metabolic reprogramming and dilated cardiomyopathy

Elizabeth D. Luczak [1]✉, Yuejin Wu[1], Jonathan M. Granger[1], Mei-ling A. Joiner[2], Nicholas R. Wilson[1], Ashish Gupta[1], Priya Umapathi[1], Kevin R. Murphy [1], Oscar E. Reyes Gaido [1], Amin Sabet[1], Eleonora Corradini[3], Wen-Wei Tseng[4], Yibin Wang [5], Albert J. R. Heck [3], An-Chi Wei [1,4]✉, Robert G. Weiss[1] & Mark E. Anderson [1]✉

Despite the clear association between myocardial injury, heart failure and depressed myocardial energetics, little is known about upstream signals responsible for remodeling myocardial metabolism after pathological stress. Here, we report increased mitochondrial calmodulin kinase II (CaMKII) activation and left ventricular dilation in mice one week after myocardial infarction (MI) surgery. By contrast, mice with genetic mitochondrial CaMKII inhibition are protected from left ventricular dilation and dysfunction after MI. Mice with myocardial and mitochondrial CaMKII overexpression (mtCaMKII) have severe dilated cardiomyopathy and decreased ATP that causes elevated cytoplasmic resting (diastolic) $Ca^{2+}$ concentration and reduced mechanical performance. We map a metabolic pathway that rescues disease phenotypes in mtCaMKII mice, providing insights into physiological and pathological metabolic consequences of CaMKII signaling in mitochondria. Our findings suggest myocardial dilation, a disease phenotype lacking specific therapies, can be prevented by targeted replacement of mitochondrial creatine kinase or mitochondrial-targeted CaMKII inhibition.

[1] Department of Medicine, The Johns Hopkins University School of Medicine, Baltimore, MD, USA. [2] Department of Internal Medicine, University of Iowa Carver College of Medicine, Iowa City, IA, USA. [3] Biomolecular Mass Spectrometry and Proteomics, Bijvoet Center for Biomolecular Research and Utrecht Institute for Pharmaceutical Sciences, Utrecht University, Utrecht, The Netherlands. [4] Department of Electrical Engineering, Graduate Institute of Biomedical Electronics and Bioinformatics, National Taiwan University, Taipei, Taiwan. [5] Departments of Anesthesiology, Physiology and Medicine, David Geffen School of Medicine, University of California, Los Angeles, CA, USA. ✉email: Betsy.Luczak@jhmi.edu; acwei86@ntu.edu.tw; Mark.Anderson@jhmi.edu

Heart failure is a leading cause of death worldwide, and failing myocardium is marked by energetic defects[1,2]. Loss of creatine kinase[3] and decreased mitochondrial respiration[4] are common findings in myocardium after pathological injury, including myocardial infarction (MI), a major cause of heart failure. Although failing hearts have reduced energy reserves, little is known about upstream control points contributing to decreased energy metabolism, nor the specific structural and functional consequences of these defects on myocardial responses to injury. MI triggers a complex set of changes in myocardium, including cardiomyocyte death and cardiac fibrosis, hypertrophy, and chamber dilation. Collectively these changes are termed adverse structural remodeling. Much is known about cellular signals contributing to myocardial hypertrophy, fibrosis, and myocyte death[5–7], whereas relatively little is known about molecular signals causing myocardial dilation[8]. Although hypertrophy has been proposed to be a precondition for myocardial chamber dilation[9], the connection between myocardial hypertrophy and cardiac chamber dilation is uncertain. The nature of the relationship between myocardial hypertrophy and cardiac chamber dilation is important clinically because current therapies are inadequate for treating or preventing pathological cardiac chamber dilation. Emergent evidence indicates that defects in mitochondrial function and energy depletion may contribute to heart failure and dilated cardiomyopathy[10].

The multifunctional $Ca^{2+}$ and calmodulin dependent protein kinase II (CaMKII) is a pluripotent signal for promoting cardiomyopathy[11]. Mouse models of myocardial CaMKII overexpression, without subcellular targeting, show activation of myocardial hypertrophy gene programs, increased cell death, fibrosis, chamber dilation, and premature death[12,13]. Excessive CaMKII activity causes myocardial hypertrophy by catalyzing phosphorylation of HDAC4, leading to cytoplasmic partitioning of HDAC4, and derepression of hypertrophic transcriptional programs[14,15]. CaMKII promotes myocardial death via multiple pathways[16], including induction of pro-apoptotic genes through NF-κB activation[17,18], defective DNA repair[19], promoting cytosolic $Ca^{2+}$ overload[20], and triggering mitochondrial permeability transition pore opening[21]. CaMKII also promotes dilated cardiomyopathy[13,22], but the mechanism is unknown. CaMKII is present in the mitochondrial fraction of heart and vascular lysates[23,24], and is found in the mitoplast fraction[25], but responses to sustained, pathological mitochondrial activation in heart are unknown.

Here, we identify increased mitochondrial CaMKII activation in failing mouse hearts 1 week after myocardial infarction surgery, and find that mouse hearts with myocardial and mitochondrial CaMKII inhibition, due to transgenic expression of a potent, and selective CaMKII inhibitor polypeptide (mtCaMKIIN)[25], are protected from left ventricular dilation and dysfunction 1 week after myocardial infarction. We develop a genetic mouse model with myocardial- and mitochondrial-targeted CaMKII overexpression, and find that excessive mitochondrial CaMKII causes dilated cardiomyopathy, without myocardial hypertrophy or death, by reducing expression of assembled complex I and the mitochondrial isoform of creatine kinase (CKmito). Genetic replacement of CKmito but not cytosolic myofibrillar creatine kinase (CK-M) is sufficient to rescue myocardial energetics, restore intracellular $Ca^{2+}$ homeostasis, and prevent dilated cardiomyopathy. Mitochondrial CaMKII overexpression coherently affects TCA cycle dehydrogenases, causing increased phosphorylation, activity and heightened production of NADH, an essential electron donor for oxidative phosphorylation. These findings show excessive mitochondrial CaMKII can trigger dilated cardiomyopathy, independent of myocardial hypertrophy and cell death, by depressing energetics.

Our findings suggest that mitochondrial CaMKII selectively contributes to cardiac chamber dilation, potentially in acquired forms of myocardial injury with high importance to public health, and that dilated cardiomyopathy can be prevented by therapies capable of restoring mitochondrial energy metabolism.

## Results

**Mitochondrial CaMKII activity increases after MI.** Myocardial infarction is a major cause of heart failure, one of the largest, unsolved public health challenges[26,27]. Myocardial CaMKII activity is increased in myocardial lysates from heart failure patients, and in many animal models of heart failure[28]. Increased CaMKII is known to contribute to myocardial hypertrophy and arrhythmias by phosphorylation of nuclear and cytoplasmic target proteins[29–31]. CaMKII was recently identified in mitochondria[23,25,32], and mitochondrial CaMKII inhibition protects against myocardial death, acutely, during the first hours after myocardial infarction[25]. However, the potential for excessive, chronic, mitochondrial CaMKII to contribute to specific myocardial disease phenotypes is unexplored. As a first step, we tested whether mitochondrial CaMKII activity is increased in failing hearts by measuring total and threonine 287 auto-phosphorylated CaMKII, a marker of CaMKII activation[33], in mitochondria isolated from mouse hearts following myocardial infarction or sham surgery. We found an increase in auto-phosphorylated CaMKII in mitochondria isolated from heart tissue 1 week after myocardial infarction surgery, compared to sham-operated controls (Fig. 1a, b). The total amount of CaMKII in the mitochondria was similar between the myocardial infarcted and sham hearts. We also observed an increase in phosphorylated CaMKII in the cytosol from the same hearts (Supplementary Fig. 1a). In addition, we quantified phosphorylation of a known mitochondrial target of CaMKII, the mitochondrial $Ca^{2+}$ uniporter (MCU)[25,32]. Rapid mitochondrial matrix $Ca^{2+}$ entry is mediated by MCU[34,35], and CaMKII increases MCU $Ca^{2+}$ entry[25,32], although this finding has been disputed[36–38]. We used recently validated phospho-specific antibodies against MCU (pMCU)[32], predicted CaMKII phosphorylation sites, and found an increase in phosphorylation of this CaMKII target in mitochondria from hearts after MI compared to sham-operated hearts (Fig. 1a, b). Collectively, these results suggest that increased mitochondrial CaMKII activity occurs in response to myocardial infarction, and persists substantially beyond the perioperative episode.

**mtCaMKIIN protects against adverse remodeling after MI.** Given the increased mitochondrial CaMKII activity in mice after myocardial infarction, we asked whether inhibition of mitochondrial CaMKII, by myocardial targeted overexpression of CaMKIIN (mtCaMKIIN)[25], could protect the heart from chronic responses to injury. We next measured mtCaMKIIN expression and found it was present in mitochondrial and extra-mitochondrial fractions (Supplementary Fig. 1b). CaMKIIN is expressed endogenously in neurons, but not in heart[39], and mtCaMKIIN hearts are protected against mitochondrial $Ca^{2+}$ overload, and increased myocardial death in the first hours after pathological stress[25]. Here, we subjected mtCaMKIIN mice and WT littermates to surgical MI, and measured heart size and function using echocardiography over 3 weeks. Although basal myocardial function, prior to MI surgery (Fig. 1c), and infarcted myocardial area 24 h after myocardial infarction surgery (Supplementary Fig. 1c) were similar between mtCaMKIIN and WT littermate controls, the WT mice showed worse left ventricular ejection fractions, and progressed to more severe left ventricular dilatation than the mtCaMKIIN mice by 1 week after myocardial infarction. The differences in left ventricular dilation and function

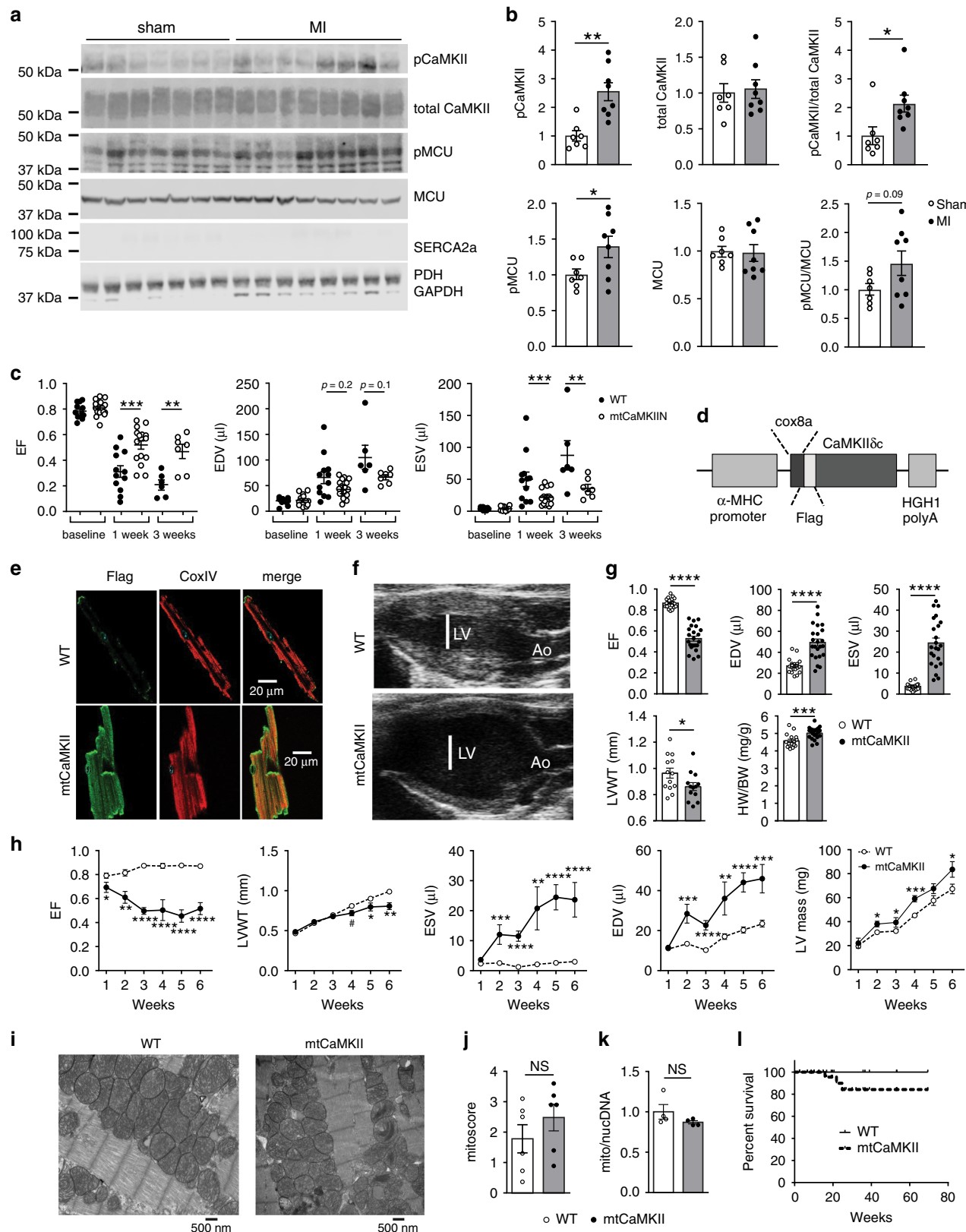

between WT and mtCaMKIIN hearts were sustained for 3 weeks after surgery, suggesting mitochondrial CaMKII participated in structural remodeling on a subacute or chronic timescale (Fig. 1c). In order to verify that mitochondrial CaMKII activity was decreased in the mtCaMKIIN hearts, we quantified pMCU and found a trend for reduced phosphorylation of this CaMKII

target 1 week following MI (Supplementary Fig. 1d). These data further support the role of mitochondrial CaMKII in structural remodeling following cardiac injury.

CaMKII has been shown to contribute to cardiac inflammation following ischemic injury[17,18,40]. In order to determine if inhibition of mitochondrial CaMKII reduces inflammation after

**Fig. 1 Mitochondrial CaMKII is activated by MI and causes dilated cardiomyopathy. a** Western blot of mitochondrial lysates and **b** summary data for phosphorylated CaMKII and total CaMKII and phosphorylated MCU and total MCU normalized to Coomassie staining in mitochondrial from sham ($n = 7$) and MI ($n = 8$) hearts. Blots for cellular compartment markers also included: SERCA2a (SR membrane), PDH (mitochondrial matrix), GAPDH (cytosol). **c** Summary data from echocardiographic measurements from WT and mtCaMKIIN mice before (WT $n = 12$, mtCaMKIIN $n = 16$), 1 (WT $n = 12$, mtCaMKIIN $n = 16$) and 3 weeks (WT $n = 6$, mtCaMKIIN $n = 7$) after MI. EF (left ventricular ejection fraction), EDV (left ventricular end-diastolic volume), ESV (left ventricular end-systolic volume). **d** Schematic of the mtCaMKII transgene construct, α-MHC: α-myosin heavy chain, cox8a: cox8a mitochondrial localization sequence, Flag: flag epitope tag, HGH1 polyA: human growth hormone polyA signal. **e** Immunofluorescence micrographs of isolated ventricular myocytes from WT and mtCaMKII mice. This experiment was repeated independently twice with similar results. **f** Representative echocardiographic images in end diastole, LV: left ventricle, Ao: aorta. **g** Summary data for echocardiographic measurements from adult WT ($n = 20$ EF, EDV, ESV; $n = 13$ LVWT; $n = 16$ HW/BW) and mtCaMKII ($n = 23$ EF, EDV, ESV; $n = 13$ LVWT; $n = 22$ HW/BW) mice. LVWT (left ventricular wall thickness), HW/BW (heart weight/body weight). **h** Summary data for echocardiographic measurements from young (1–6 weeks of age) WT ($n = 12$) and mtCaMKII ($n = 6$) mice. **i** Representative transmission electron micrographs from WT and mtCaMKII left ventricle. **j** Mitochondrial injury score for WT ($n = 6$) and mtCaMKII ($n = 6$) hearts. **k** Mitochondrial DNA content (cox1) normalized to nuclear DNA (βglobin) for WT ($n = 4$) and mtCaMKII ($n = 4$) hearts. **l** Kaplan–Meier survival relationship for WT ($n = 41$) and mtCaMKII ($n = 48$) mice. Data are represented as mean ± SEM, significance was determined using a two-tailed Student's $t$ test or log-rank test (survivorship). ****$P < 0.0001$, ***$P < 0.001$, **$P < 0.01$, *$P < 0.05$, NS = not significant. Source data, including exact $p$ values, are provided as a Source data file.

MI, we measured chemokine expression and leukocyte infiltration 1 week after coronary ligation. We did find evidence of increased inflammation after MI surgery, but found no difference in CCL2 or CCL3 mRNA expression (Supplementary Fig. 1e), nor CD45+ cell infiltration (Supplementary Fig. 1f) in mtCaMKIIN hearts compared to WT littermates. These data suggest that inhibition of mitochondrial CaMKII does not affect the inflammatory response in this model of injury.

**Increased mitochondrial CaMKII causes dilated cardiomyopathy.** To directly test the effects of increased CaMKII activity in cardiac mitochondria, we developed a mouse model with chronic transgenic overexpression of CaMKII targeted to myocardial mitochondrial matrix (mtCaMKII). We fused the CaMKIIδ cDNA to a Cox8a mitochondrial localization sequence and a flag epitope tag, and expressed this construct under the control of the α-myosin heavy chain promoter[41] (Fig. 1d). Cardiac myocytes isolated from mtCaMKII mice showed flag-CaMKII localized primarily with a mitochondrial marker (CoxIV) (Fig. 1e). The mtCaMKII mice developed dilated cardiomyopathy with reduced left ventricular ejection fraction, increased left ventricular volumes, and decreased left ventricular wall thickness (Fig. 1f, g). We observed modest cardiac hypertrophy in the mtCaMKII mice compared to WT littermates, as measured by heart weight normalized to body weight (Fig. 1g, non-normalized data shown in Supplementary Fig. 2a), but cardiac myocyte cross-sectional area was not different in mtCaMKII hearts compared to WT littermate hearts (Supplementary Fig. 2b, c). Left ventricular dilation was apparent by 2 weeks of age, and preceded detectable hypertrophy in mtCaMKII hearts, compared to WT littermate controls (Fig. 1h). Mitochondrial ultrastructural integrity scoring[25] (Fig. 1i, j), and abundance, reflected by the ratio of mitochondrial to nuclear DNA (Fig. 1k), were similar compared to WT littermate controls. These data show that mtCaMKII mice develop a severe dilated cardiomyopathy with minimal myocardial hypertrophy.

**Mitochondrial CaMKII does not affect myocardial hypertrophy.** In contrast to the dilated cardiomyopathy and near normal lifespan in mtCaMKII mice (Fig. 1l), myocardial CaMKII overexpression in the absence of subcellular targeting causes marked cardiac hypertrophy, left ventricular dilation, increased myocardial cell death, cardiac fibrosis, heart failure, and premature death[12,13]. The striking differences in the phenotypes of mtCaMKII, and mice with transgenic myocardial CaMKII overexpression lacking subcellular targeting appeared to confirm a

specific role for mitochondrial CaMKII in dilated cardiomyopathy. However, while the Cox8a was effective in predominantly targeting CaMKII overexpression to myocardial mitochondria, we also detected small amounts of mtCaMKII in extra-mitochondrial compartments (Fig. 2a). To determine if extra-mitochondrial mtCaMKII contributed to cardiomyopathy in mtCaMKII mice, we crossed the mtCaMKII mice with mice expressing AC3-I, a CaMKII inhibitory peptide that is fused to enhanced green fluorescent protein (GFP)[42], and excluded from mitochondria (Fig. 2b). The double transgenic mice, combining mitochondrial-targeted CaMKII overexpression and extra-mitochondrial CaMKII inhibition, showed persistent left ventricular dysfunction (see left ventricular ejection fraction) and dilation (see left ventricular end-systolic and diastolic volumes), but loss of left ventricular hypertrophy (see heart weight/body weight), compared to the mtCaMKII mice (Fig. 2c). The partial improvement of left ventricular ejection fraction in the mtCaMKII x AC3-I interbred mice suggested that mitochondrial and extramitochondrial CaMKII each have the potential to decrease myocardial function, but the unchanged end-diastolic volumes indicate that mitochondrial CaMKII exclusively promotes dilated cardiomyopathy, and not myocardial hypertrophy. We next crossed the mtCaMKII mice with mtCaMKIIN mice[25]. The coexpression of mtCaMKIIN reversed the dilated cardiomyopathy phenotype in the mtCaMKII mice (Fig. 2d), without reducing mtCaMKII expression (Fig. 2e, f), confirming that dilated cardiomyopathy is due to excessive mitochondrial CaMKII activity, and excluding the possibility that dilated cardiomyopathy is a non-specific consequence of mitochondrial-targeted transgenic protein overexpression. Considered together with previously published information[13], these data suggest that the dilated phenotype of the mtCaMKII heart is due to increased CaMKII activity in mitochondria, while myocardial hypertrophy and premature death are predominantly related to extra-mitochondrial actions of CaMKII.

**Normal mitochondrial Ca²⁺ in mtCaMKII hearts.** CaMKII plays a prominent role in myocardial biology, in part, by enhancing activity of intracellular $Ca^{2+}$ homeostatic proteins[11]. Mitochondrial-targeted CaMKII inhibition, in mtCaMKIIN mice, protects against acute responses to myocardial injury by reducing cardiomyocyte death, $Ca^{2+}$ overload, and loss of $\Delta\Psi_{mito}$ (ref. [25]). Mitochondria isolated from the hearts of mtCaMKII mice showed significantly increased pMCU (Fig. 2g, h), suggesting excess phosphorylation at these sites in mtCaMKII compared to WT MCU. Based on these findings, we initially anticipated that

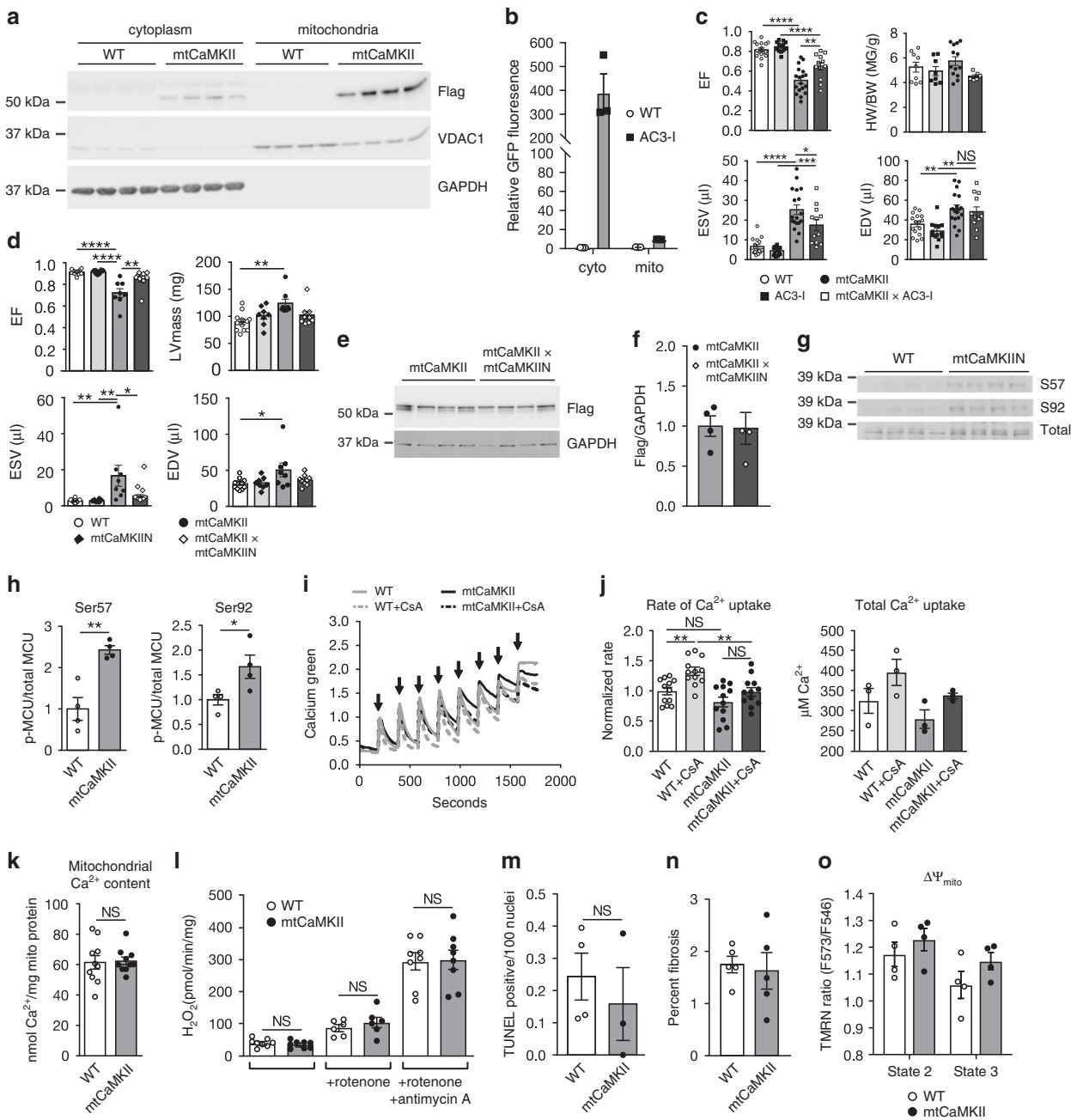

**Fig. 2 mtCaMKII does not affect hypertrophy, MCU, mitochondrial Ca²⁺, ROS, nor ΔΨ_mito. a** Western blot for flag-CaMKII, VDAC1, and GAPDH in cytoplasm and mitochondria fractions from WT ($n = 4$) and mtCaMKII hearts ($n = 4$). **b** Quantification AC3-I-GFP fusion protein expression in cytoplasm and mitochondria from WT and AC3-I mouse hearts ($n = 3$). **c** Summary data for echocardiographic measurements from WT ($n = 16$), AC3-I ($n = 13$), mtCaMKII ($n = 18$), and mtCaMKII x AC3-I ($n = 12$) mice. **d** Summarized echocardiographic measurement data from WT ($n = 12$), mtCaMKIIN ($n = 8$), mtCaMKII ($n = 10$), and mtCaMKII x mtCaMKIIN ($n = 10$) mice. **e** Western blot and **f** summary data for flag-CaMKII normalized to GAPDH in mtCaMKII ($n = 4$) and mtCaMKII x mtCaMKIIN ($n = 4$) hearts. **g** Western blot and **h** summary data for phosphorylated MCU normalized to total MCU in mitochondria from WT ($n = 4$) and mtCaMKII ($n = 4$) hearts. **i** Mitochondrial Ca²⁺ uptake assay with cell membrane permeabilized adult ventricular myocytes from WT ($n = 3$) and mtCaMKII ($n = 3$) mice; arrows indicate addition of Ca²⁺. **j** Rate of Ca²⁺ uptake in permeabilized myocytes calculated from the first 3 peaks of Ca²⁺ addition and total Ca²⁺ uptake before mPTP opening in WT ($n = 3$) and mtCaMKII ($n = 3$) mice. **k** Total mitochondrial Ca²⁺ content normalized to total protein in isolated mitochondria from WT ($n = 10$) and mtCaMKII ($n = 10$) hearts. **l** H₂O₂ production measured by Amplex Red in isolated mitochondria from WT ($n = 8$) and mtCaMKII ($n = 8$) hearts. **m** Quantification of TUNEL positive nuclei from heart sections of WT (5 sections from $n = 4$ hearts) and mtCaMKII (5 sections from $n = 3$ hearts) mice. **n** Quantification of fibrosis by Masson's Trichrome staining from WT (6 sections from $n = 3$ hearts) and mtCaMKII (6 sections from $n = 3$ hearts) heart sections. **o** Mitochondrial membrane potential (ΔΨ_mito) measured with TMRM under state 2 (substrate alone) and state 3 (substrate plus ADP) respiration in isolated mitochondria from WT ($n = 4$) and mtCaMKII ($n = 4$) hearts. Data are represented as mean ± SEM, significance was determined using a two-tailed Student's $t$ test or one-way ANOVA with Tukey's multiple comparisons test. ****$P < 0.0001$, ***$P < 0.001$, **$P < 0.01$, *$P < 0.05$, NS = not significant. Source data, including exact $p$ values, are provided as a Source data file.

dilated cardiomyopathy in mtCaMKII mice was related to mitochondrial $Ca^{2+}$ overload, loss of $\Delta\Psi_{mito}$, and increased myocardial death.

To test if mitochondrial $Ca^{2+}$ entry was enhanced in mtCaMKII mitochondria, we measured MCU-mediated mitochondrial $Ca^{2+}$ uptake, and mitochondrial $Ca^{2+}$ content. We found that mitochondria from mtCaMKII and WT littermate control hearts had similar $Ca^{2+}$ uptake rates (Fig. 2i, j), and no differences in mitochondrial $Ca^{2+}$ content (Fig. 2k). In order to determine whether the mitochondrial transition pore contributed to the apparent rate of mitochondrial $Ca^{2+}$ uptake, we also included cyclosporine A (CsA), an inhibitor of the mPTP, in the $Ca^{2+}$ uptake assays (Fig. 2i, j). We found an increase in the apparent rate of $Ca^{2+}$ uptake in WT mitochondria, and a modest reduction in apparent rate of $Ca^{2+}$ uptake in mtCaMKII compared to WT mitochondria in the presence of CsA, suggesting a reduction in mPTP activity in the mtCaMKII mitochondria. Taken together, these studies supported a view contrary to our starting hypothesis: that chronic mitochondrial CaMKII overexpression did not cause dilated cardiomyopathy by disrupting mitochondrial $Ca^{2+}$ homeostasis. Furthermore, mtCaMKII mice did not exhibit increased mitochondrial ROS (Fig. 2l), cardiomyocyte TUNEL staining (Fig. 2m), or fibrosis (Fig. 2n), and had similar $\Delta\Psi_{mito}$ compared to WT littermate controls (Fig. 2o). Measuring mitochondrial ROS is technically challenging[43], so we considered the possibility that our studies were inadequate to detect differences in ROS between WT and mtCaMKII hearts, potentially of a magnitude to contribute to the dilated cardiomyopathy in mtCaMKII mice. In order to more decisively determine if elevated ROS contributed to mtCaMKII cardiomyopathy, we took an orthogonal approach, and interbred mtCaMKII mice with mice transgenically overexpressing a mitochondrial-targeted form of catalase (mCat), an enzyme that decomposes $H_2O_2$ into $H_2O$ and $O_2$. The mCat mice are resistant to ROS mediated mitochondrial disease[44]. However, the mtCaMKII x mCat interbred mice were not protected against left ventricular dilation, nor depressed ejection fraction compared to mtCaMKII mice (Supplementary Fig. 3a). Finally, Nnt is mutated in the C57BL/6J strain we used[45], suggesting that the left ventricular dilation in mtCaMKII cardiomyopathy phenotype may be artificially disconnected from ROS generation[46]. In order to examine this possibility, we back crossed mtCaMKII mice for five generations into the CD1 background that expresses WT Nnt. The mtCaMKII mice in the C57Bl/6J and CD1 genetic backgrounds exhibited similar left ventricular dilation (Supplementary Fig. 3b), suggesting lack of increased ROS generation in mtCaMKII hearts was not due to loss of Nnt. Thus, in vitro and in vivo data strongly suggested that mtCaMKII cardiomyopathy was not a consequence of elevated ROS. Given the lack of effect of increased mitochondrial CaMKII activity on myocyte survival, mitochondrial $Ca^{2+}$, ROS, or $\Delta\Psi_{mito}$, activators and measures of the mitochondrial transition pore (mPTP) opening, we did not anticipate mPTP was involved in the dilated cardiomyopathy present in mtCaMKII hearts. However, to directly test for increased mPTP opening in mtCaMKII hearts in vivo we interbred mtCaMKII mice with mice lacking cyclophilin D ($Ppif^{-/-}$), an mPTP subunit that is involved in pore opening[47]. The $Ppif^{-/-}$ mice are protected from mPTP opening[48]. Consistent with our data up to this point, the $Ppif^{-/-}$ x mtCaMKII interbred mice were not protected against dilated cardiomyopathy (Supplementary Fig. 3c). Taken together, these in vitro and in vivo results strongly argued against defective mitochondrial $Ca^{2+}$ or its consequences as a cause of cardiomyopathy in mtCaMKII mice.

### Impaired cytoplasmic $Ca^{2+}$ homeostasis in mtCaMKII hearts.
Energy deficiency may contribute to dilated cardiomyopathy[49], but little is known about specific upstream molecular signals initiating metabolic insufficiency. We initially found that the ATP concentration was significantly reduced in myocardial mitochondria lysates from mtCaMKII compared to WT littermate control hearts (Fig. 3a). We next used $^{31}P$ magnetic resonance (MR) spectroscopy to measure myocardial high-energy phosphates in vivo[50,51]. Cardiac ATP and creatine phosphate concentrations were significantly reduced, by 30–45%, in mtCaMKII mice as compared to WT controls (Fig. 3b, c). MR imaging confirmed dilated cardiomyopathy (Fig. 3d, Supplementary Movie 1), similar to our echocardiographic findings (Fig. 1g). Together, these results show a significant reduction in ATP in mtCaMKII hearts, in vitro and in vivo, and are consistent with the hypothesis that impaired myocardial function in mtCaMKII mice is a consequence of energy deficiency.

Cytoplasmic $Ca^{2+}$ concentration ($[Ca^{2+}]_i$) controls cardiac contraction and relaxation, and $[Ca^{2+}]_i$ homeostasis represents a major physiological energy cost. We next measured $[Ca^{2+}]_i$ in ventricular myocytes isolated from mtCaMKII and WT hearts in response to increasing stimulation frequencies. Contrary to our expectations, these initial studies showed a decrease in resting $[Ca^{2+}]_i$ in cardiomyocytes isolated from mtCaMKII and WT hearts (Supplementary Fig. 4a) consistent with previous findings in hearts with CaMKII overexpression[52]. We next considered that increased cytoplasmic CaMKII seen in mtCaMKII cardiomyocytes (Fig. 2a) could mask effects of ATP deficiency on $[Ca^{2+}]_i$ in isolated, mechanically unloaded cardiomyocytes, potentially because energy requirements were too low to reveal differences. In order to test this idea, we repeated the $[Ca^{2+}]_i$ measurements in isolated ventricular myocytes dialyzed with AIP, a CaMKII inhibitory peptide. The AIP peptide was confined to the cytoplasmic compartment because it lacked a membrane permeation epitope and was dialyzed across the cell membrane using a microelectrode in whole-cell mode patch-clamp configuration. AIP dialysis had no effect on resting diastolic $[Ca^{2+}]_i$ in WT cardiomyocytes, but resulted in a significant, stimulation frequency dependent elevation in diastolic $[Ca^{2+}]_i$ (Fig. 3e, f), unchanged peak $Ca^{2+}$ amplitude (Supplementary Fig. 4b) and slower $Ca^{2+}$ decay (Supplementary Fig. 4c) in mtCaMKII cardiomyocytes, consistent with the hypothesis that incompletely targeted mtCaMKII preserved cytoplasmic $[Ca^{2+}]$ homeostasis in ventricular myocytes isolated from mtCaMKII hearts. In order to test if the elevated $[Ca^{2+}]_i$ in mtCaMKII cardiomyocytes was a consequence of deficient ATP, we fortified the micropipette dialysate with ATP (5 mM) to approximate physiological cytoplasmic ATP activity[51]. ATP dialysis reduced diastolic $[Ca^{2+}]_i$ in mtCaMKII cardiomyocytes to WT levels, but had no effect on diastolic $[Ca^{2+}]_i$ in WT control cardiomyocytes (Fig. 3e, f). We interpreted the data up to this point as supporting a model where impaired energetics in mtCaMKII hearts caused defective intracellular $[Ca^{2+}]$ homeostasis, a cardinal feature of myocardial dysfunction.

### CKmito prevents dilated cardiomyopathy in mtCaMKII mice.
Loss of creatine kinase activity is a common finding in failing human myocardium and in animal models of heart failure[53,54]. Creatine kinase maintains energetics by rapidly and reversibly transferring a high-energy phosphoryl moiety between ATP and creatine phosphate, a highly diffusible high-energy phosphate molecule important for cellular energy homeostasis. Overexpression of CK-M, the cytosolic myofibrillar isoform of creatine kinase, in several models of heart failure improves overall cardiac function and survival[55,56], but the potential role of CKmito, the mitochondrial isoform, was only recently shown to improve myocardial response to injury. We considered that loss of CKmito could be important to the dilated cardiomyopathy in mtCaMKII

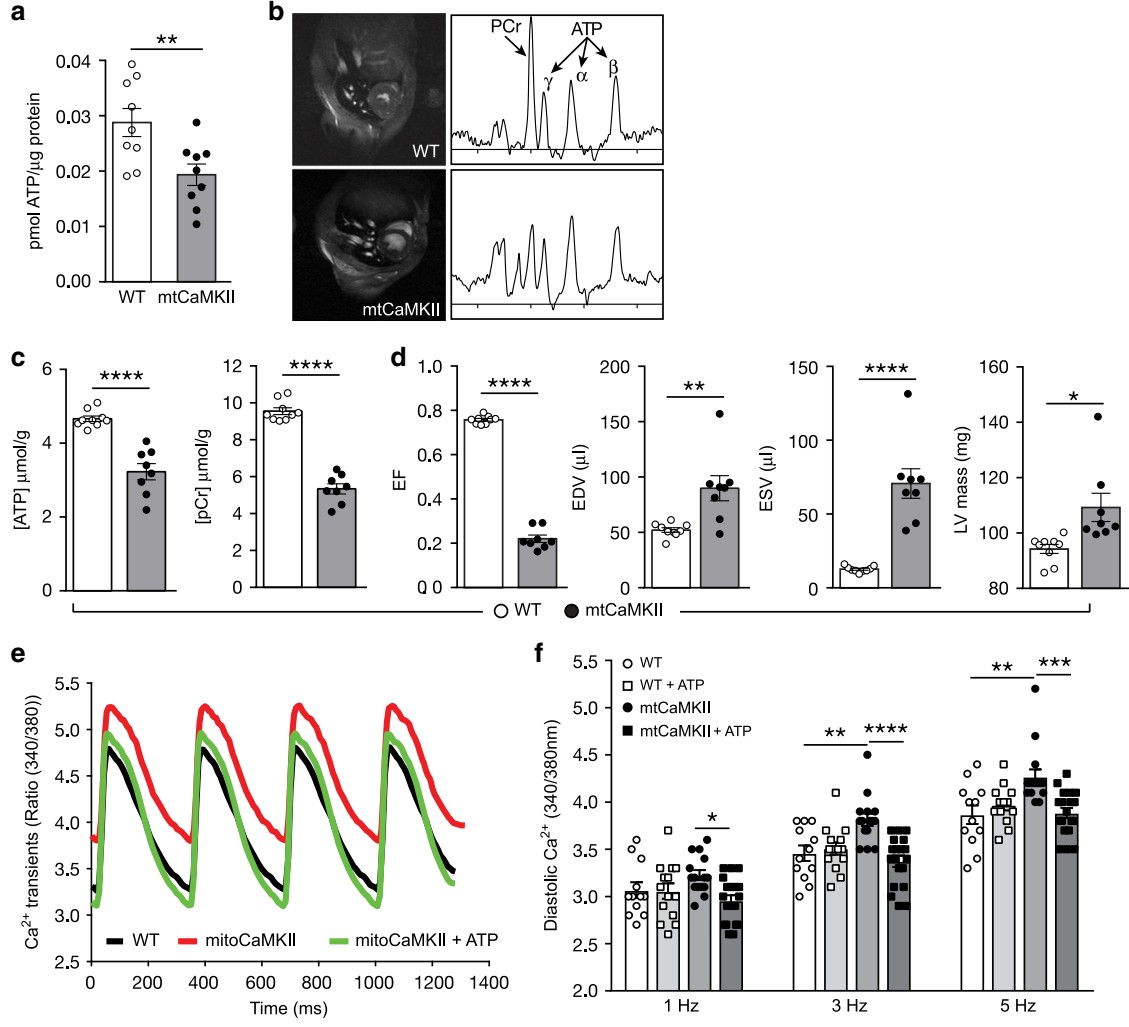

**Fig. 3 mtCaMKII hearts are ATP deficient. a** ATP content in mitochondria isolated from WT ($n = 9$) and mtCaMKII ($n = 9$) hearts. **b** Example MRI images and corresponding ³¹P MR spectra and **c** summary data for ATP and creatine phosphate (PCr) quantification from in vivo ³¹P MR spectroscopy in WT ($n = 9$) and mtCaMKII ($n = 8$) hearts. **d** Summary data from MRI measurements of WT ($n = 9$) and mtCaMKII ($n = 8$) hearts; see Supplementary Movie 1 for MR Cines. **e** Example intracellular [Ca²⁺] signals recorded from Fura-2 and **f** summary data for diastolic (resting) [Ca²⁺] from isolated ventricular myocytes stimulated at 1, 3, or 5 Hz (WT $n = 12$; WT+ATP $n = 14$; mtCaMKII $n = 15$; mtCaMKII+ATP $n = 19$) (WT cells were isolated from 2 heart, mtCaMKII cells were isolated from 3 hearts). Data are represented as mean ± SEM, significance was determined using a two-tailed Student's $t$ test or one-way ANOVA with Tukey's multiple comparisons test. ****$P < 0.0001$, ***$P < 0.001$, **$P < 0.01$, *$P < 0.05$. Source data, including exact $p$ values, are provided as a Source data file.

mice, as it associates with the ATP transporter on the mitochondrial intermembrane space, and enhances energy transfer from the mitochondria to the cytosol[57]. Furthermore, CKmito was hyperphosphorylated in mtCaMKII compared to WT mitochondria (KCRS, Supplementary Table 1). We measured expression of myofibrillar, and mitochondrial creatine kinase isoforms in WT and mtCaMKII hearts with antibodies we validated to be specific for each isoform location (Supplementary Fig. 5a), and found a trend ($P = 0.06$) for reduced CKmito in mtCaMKII compared to WT (Fig. 4a, b). In contrast, the expression of CK-M was similar in WT and mtCaMKII hearts (Fig. 4c, d). However, we did not measure a reduction in CKmito 1 week after MI surgery compared to sham-operated WT mice (Supplementary Fig. 5b), suggesting mitochondrial CaMKII overexpression in mtCaMKII hearts is a more potent upstream signal for reducing CKmito than MI under these conditions. In order to test whether ATP loss and reduced CKmito contributed to the dilated phenotype in the mtCaMKII hearts, we attempted to recover myocardial energetics by interbreeding mtCaMKII

mice with mice overexpressing CKmito in myocardium. CKmito was significantly overexpressed at a similar level in both the CKmito and mtCaMKII x CKmito interbred mice (Fig. 4e, f). The mtCaMKII x CKmito interbred mice showed improved in vivo ATP and creatine phosphate levels (Fig. 4g), reduced diastolic [Ca²⁺]ᵢ in isolated myocytes (Fig. 4h), and exhibited significantly reduced left ventricular dilation compared to mtCaMKII littermates (Fig. 4i and Supplementary Movie 1). Importantly, the recovery of myocardial function and energetics in the mtCaMKII x CKmito mice occurred without a reduction in mitochondrial CaMKII overexpression (Fig. 4j, k). In contrast, interbreeding mtCaMKII mice with transgenic mice overexpressing the cytosolic CK-M did not rescue dilated cardiomyopathy (Supplementary Fig. 5c), but rather worsened adverse structural remodeling compared to mtCaMKII hearts. These data show that defective energetics and dilated cardiomyopathy in the mtCaMKII mice can be significantly reversed by CKmito replacement. These findings provide evidence that myocardial dilation can be a metabolic consequence of excessive mitochondrial CaMKII

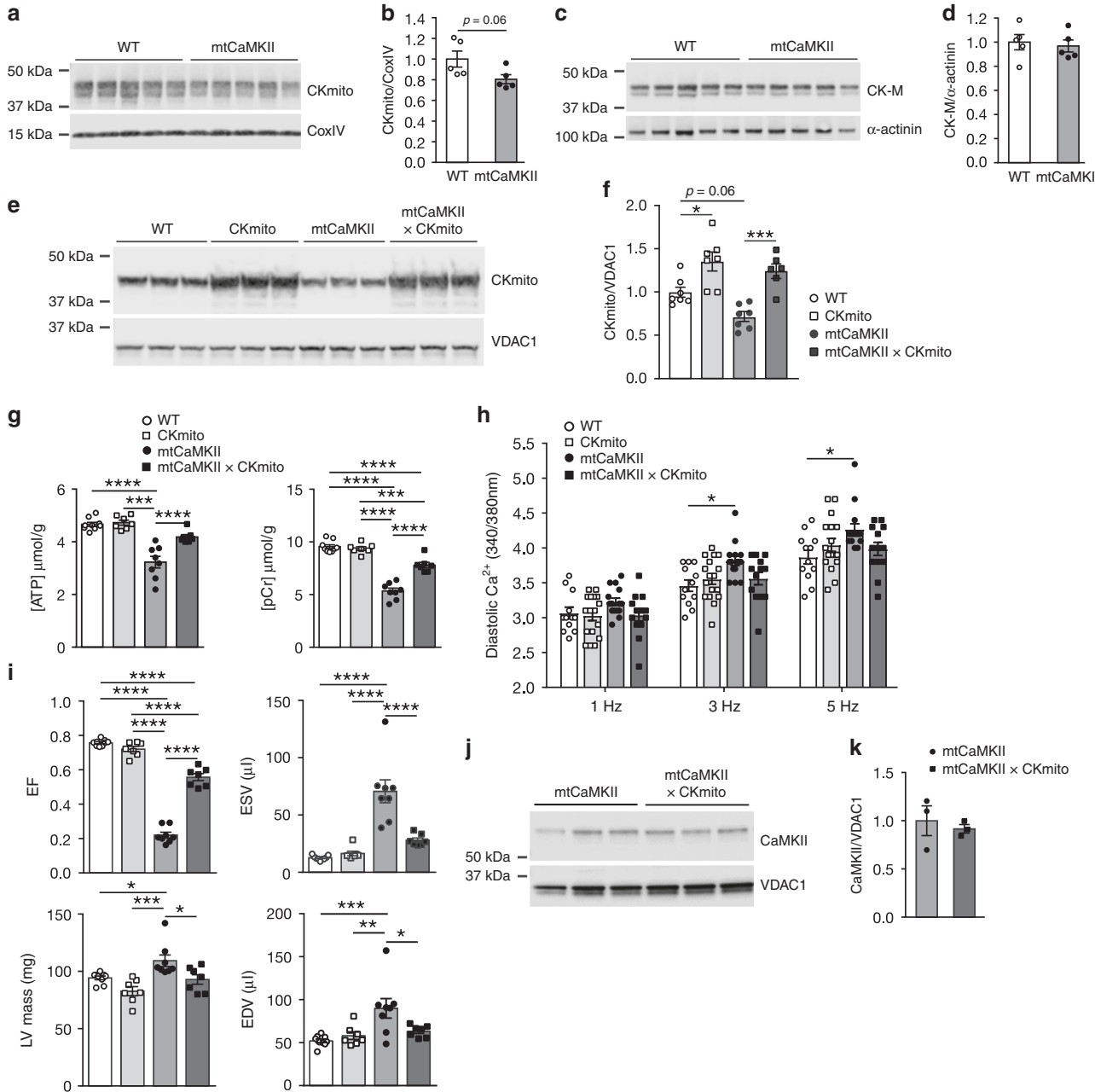

**Fig. 4 CKmito replacement rescues ATP deficiency in mtCaMKII hearts. a** Western blots and **b** summary data for CKmito expression normalized to CoxIV from WT ($n = 5$) and mtCaMKII ($n = 5$) hearts. **c** Western blots and **d** summary data for CK-M expression normalized to α-actinin from WT ($n = 5$) and mtCaMKII ($n = 5$) hearts. **e** Western blot and **f** summary data for CKmito expression normalized to VDAC1 from WT ($n = 7$), CKmito ($n = 7$), mtCaMKII ($n = 7$), and mtCaMKII x CKmito ($n = 6$) hearts. **g** In vivo ATP (left) and PCr (right) quantification from ³¹P MR spectroscopy in WT ($n = 9$), CKmito ($n = 7$), mtCaMKII ($n = 8$), and mtCaMKII x CKmito ($n = 7$) hearts. **h** Summary data for diastolic [Ca²⁺] measurements made with Fura-2-loaded ventricular myocytes stimulated at 1, 3, and 5 Hz (WT $n = 12$; CKmito $n = 17$; mtCaMKII $n = 15$; mtCaMKII x CKmito $n = 13$ ventricular myocytes isolated from 2 hearts/group). **i** Summary data from in vivo MRI measurements in WT ($n = 9$), CKmito ($n = 7$), mtCaMKII ($n = 8$), and mtCaMKII x CKmito ($n = 7$) hearts. **g–i** WT and mtCaMKII data are the same as in Fig. 3. **j** Western blot and **k** summary data for CaMKII normalized to VDAC1 in mitochondrial lysates from mtCaMKII ($n = 3$) and mtCaMKII x CKmito ($n = 3$) hearts. Data are represented as mean ± SEM, significance was determined using a two-tailed Student's t test or one-way ANOVA with Tukey's multiple comparisons test. ****$P < 0.0001$, ***$P < 0.001$, **$P < 0.01$, *$P < 0.05$. Source data, including exact p values, are provided as a Source data file.

activity leading to ATP deficiency, and suggest that chronic ATP deficiency can result in dilated cardiomyopathy that is potentially preventable or reversible.

**mtCaMKII decreases complex I expression and activity.** Loss of electron transport chain (ETC) complex components are thought

to drive reduced metabolic capacity in some genetic and acquired forms of heart failure[58,59]. With this in mind, we next evaluated the ETC to determine if reduced ATP production in mtCaMKII hearts was associated with loss of ETC complex expression and/or function. We interrogated ETC complex expression in heart lysates, and found a significant decrease in protein expression of a complex I component (NDUFB8), and a significant increase in

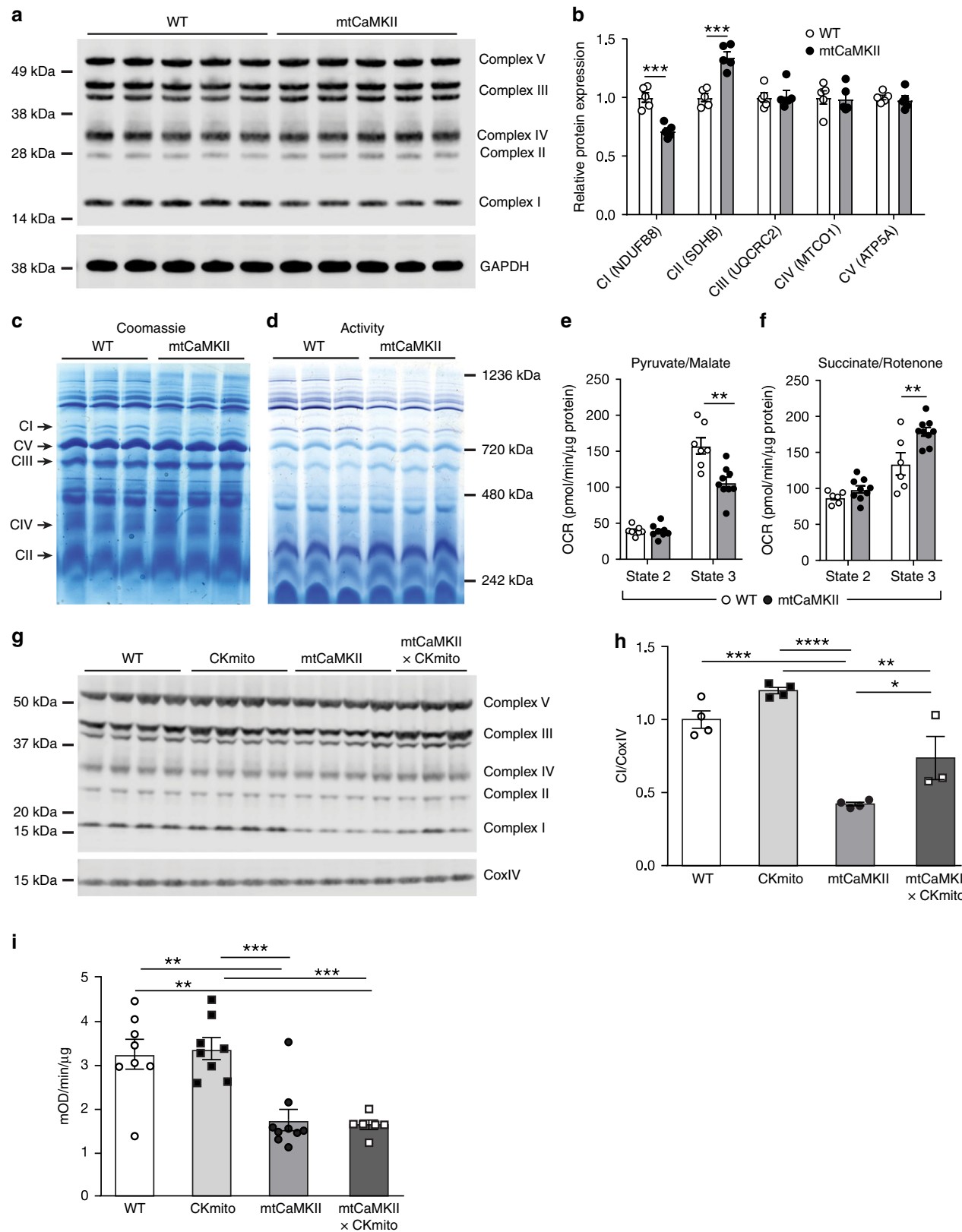

protein expression of a complex II component (SDHB) in mtCaMKII compared to WT hearts (Fig. 5a, b). Because these complexes are made up of many proteins, we quantified assembled complexes and measured complex activity using native gel electrophoresis. We found a significant decrease in assembled complex I (Fig. 5c) and in complex I activity (Fig. 5d) in

mtCaMKII mitochondria compared to WT. Based on the apparent decrease in complex I and increase in complex II in mtCaMKII hearts, we measured oxygen consumption rates of isolated mitochondria under conditions selective for complex I (pyruvate/malate) and complex II (succinate/rotenone) metabolism. We observed a decrease in state 3 (ADP activated)

**Fig. 5 mtCaMKII hearts show decreased complex I and increased complex II activity. a** Western blot of heart lysates from WT ($n = 5$) and mtCaMKII ($n = 5$) using OxPhos antibody. **b** Protein expression for complexes I–V normalized to GAPDH quantified from (**a**). **c** Blue Native gel with mitochondria from WT and mtCaMKII hearts stained with Coomassie blue and **d** in-gel activity assays for complex I. These experiments were repeated twice with similar results. **e, f** Mitochondrial respiration measured under state 2 (substrate alone) and state 3 (substrate plus ADP) conditions in isolated mitochondria from WT ($n = 7$) and mtCaMKII ($n = 9$) hearts. Substrates favored complex I (pyruvate/malate, **e**) or complex II (succinate/rotenone, **f**) activity. **g** Western blot with OxPhos antibody and (**h**) summary data for complex I normalized to CoxIV in heart lysates from WT ($n = 4$), CKmito ($n = 4$), mtCaMKII ($n = 4$), and mtCaMKII x CKmito ($n = 3$) hearts. **i** Complex I activity measured in mitochondria isolated from WT ($n = 8$), CKmito ($n = 8$), mtCaMKII ($n = 9$), and mtCaMKII x CKmito ($n = 6$) hearts. Data are represented as mean ± SEM, significance was determined using a two-tailed Student's $t$ test or one-way ANOVA with Tukey's multiple comparisons test. \*\*\*$P < 0.001$, \*\*$P < 0.01$, \*$P < 0.05$. Source data, including exact $p$ values, are provided as a Source data file.

respiration in the presence of complex I substrates (pyruvate/malate) (Fig. 5e), and an increase in state 3 respiration in the presence of complex II substrates (succinate/rotenone) (Fig. 5f) in the mtCaMKII mitochondria compared to WT, consistent with protein expression and activity data. We next repeated the measurement of complex I expression in mtCaMKII mitochondria rescued by CKmito overexpression. We found a partial recovery of expression of a complex I component in mtCaMKII x CKmito compared to mtCaMKII mitochondria (Fig. 5g, h), but no recovery of complex I activity (Fig. 5i). We interpret these data to collectively indicate that the reduction in ATP production in mtCaMKII hearts is a consequence of extensive metabolic remodeling that can be substantially corrected by a modest replacement of CKmito.

**Augmented TCA cycle dehydrogenases in mtCaMKII hearts.** To more broadly interrogate potential metabolic targets for CaMKII in mitochondria we performed a mass spectrometry-based analysis of phosphorylated peptides enriched from myocardial mitochondria from mtCaMKII and WT littermate hearts (see Methods). We identified 38 proteins exhibiting significantly increased phosphorylation in the mtCaMKII mice compared to the WT controls. Most notably, 22 of these proteins were involved in metabolism and energy production (Supplementary Table 1). The abundance of these targets, together with our finding that mtCaMKII have impaired energetics, added further support to our earlier results showing that CaMKII is involved in regulating mitochondrial metabolism.

We were intrigued by a pattern of increased phosphorylation of pyruvate dehydrogenase, and key tricarboxylic acid (TCA) cycle dehydrogenases in the mtCaMKII hearts (Supplementary Table 1). We next asked if this hyperphosphorylation was associated with a change in activity of these enzymes. The activity of pyruvate dehydrogenase (Fig. 6a), α-ketogluterate dehydrogenase, fumarase, and malate dehydrogenase (Fig. 6b) were significantly increased in mtCaMKII compared to WT mitochondria, and there was a trend toward increased activity in all TCA cycle enzymes in mtCaMKII compared to WT mitochondria. These findings suggested that mtCaMKII operated, at least in part, to augment metabolism through TCA cycle activity, possibly providing insight into acute metabolic benefits of physiological increases in mitochondrial CaMKII activity. Because the TCA cycle operates to produce NADH, the major electron source for ATP production by oxidative phosphorylation, we measured NADH in ventricular myocytes isolated from mtCaMKII and WT hearts (Fig. 6c). Consistent with augmented TCA cycle activity, resting NADH was increased in mtCaMKII compared to WT cardiomyocytes (Fig. 6d). However, pacing induced NADH increases were only present in cardiomyocytes isolated from WT mice. In contrast, increased pacing frequency caused a reduction in NADH in mtCaMKII cardiomyocytes (Fig. 6d), suggesting that the resting production of NADH was augmented, but lacked capacity to increase NADH production during rapid

stimulation. Intriguingly, and in contrast to other models of cardiomyopathy with loss of complex I, we did not detect a significant deficiency of $NAD^+$ in cardiac lysates (Fig. 6e). $NAD^+$ is an essential cofactor for sirtuins, mitochondrial deacetylases whose loss of function is linked to cardiomyopathy[60,61]. We measured an increase in mitochondrial protein acetylation (Fig. 6f, g), potentially consistent with previous reports in cardiomyopathy induced by loss of complex I[62]. However, this enhancement of acetylation was apparently inadequate to cause loss of $\Delta\Psi_{mito}$ (Fig. 2o), or increase cardiomyocyte death (Fig. 2m, n), downstream events in cardiomyopathy attributed to $NAD^+$ deficiency and increased mitochondrial protein acetylation[62]. We interpret these data to indicate excessive mitochondrial CaMKII activity causes profound metabolic remodeling, despite augmented PDH and TCA cycle activity.

**Modeling mtCaMKII and metabolism.** We performed analysis of mitochondrial metabolism using a modified computational model simulating the cardiac cycle using pulsatile functions for ATP hydrolysis and cytosolic calcium transients[63,64] (see Methods) (Fig. 7a). The effect of mtCaMKII was simulated by reduced CKmito activity and complex I respiration, and increased TCA enzyme activities, as observed in experimental measurements (Supplementary Table 2). This mathematical simulation showed decreased ATP and creatine phosphate levels in mtCaMKII, and replacement of CKmito produced a substantial rescue of these parameters (Fig. 7b and Supplementary Fig. 6), similar to our experimental observations. We interpret the consistency between these simulated and experimental outcomes to suggest that the combination of reduced CKmito and complex I activity are sufficient to cause decreased ATP, despite enhanced pyruvate dehydrogenase and TCA cycle activity, and that ATP deficiency can be repaired by CKmito replacement in the presence of excessive mtCaMKII.

**Discussion**
The association between defective ATP metabolism and heart failure has been recognized for many years[65–67], and diminished expression of creatine kinase and complex I have been noted in animal models, and in failing human hearts[54,55,68–70]. Despite this, there has been uncertainty whether failed energetics are a cause or a consequence of damaged myocardium. Thus, lack of understanding of upstream signals controlling energetics and cardiomyopathy constitutes an important knowledge gap. Our study provides evidence that mitochondrial CaMKII can orchestrate pathological metabolic remodeling and left ventricular dilation. mtCaMKII hearts are deficient in ATP (Fig. 3c), dilated and manifest contractile dysfunction and deranged $Ca^{2+}$ homeostasis, but are free of hypertrophy and fibrosis. We acknowledge that the derangement in energetics could be secondary to the severe pathology we observed in the mtCaMKII hearts since we were unable measure ATP in hearts prior to the development of

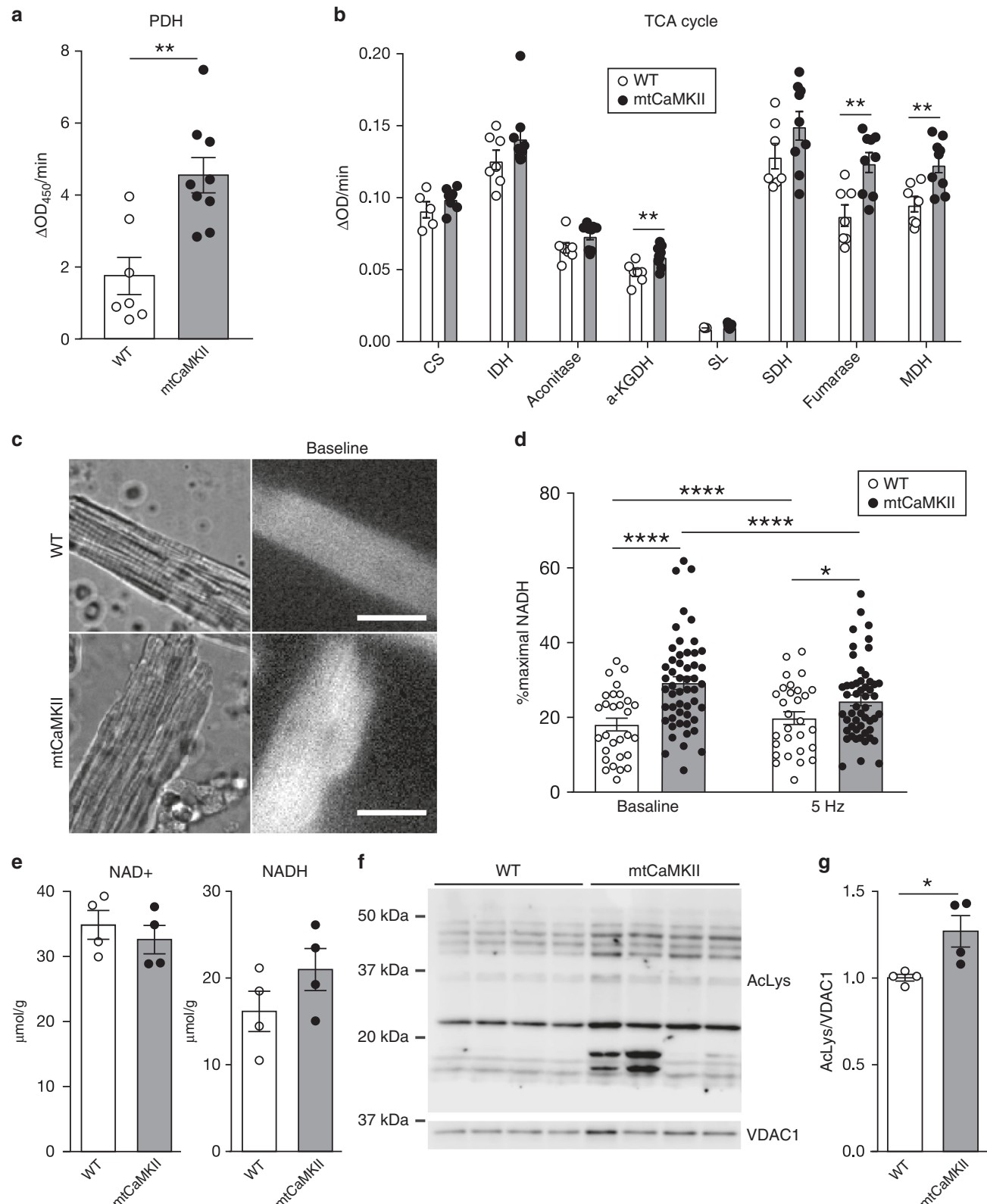

these pathological phenotypes due to technical limitations. However, excessive mitochondrial CaMKII activity is almost certainly the inciting trigger due to the nature of mitochondrial-targeted CaMKII overexpression in this model. In contrast to mice with untargeted myocardial CaMKII overexpression, the mtCaMKII do not show a high rate of premature death, despite depressed energetics and cardiac chamber dilation. Restoration of

a single enzyme, CKmito, by interbreeding mtCaMKII with CKmito transgenic mice, substantially repaired energetics and cardiac dysfunction, and restored $[Ca^{2+}]_i$ homeostasis (Fig. 4g–i). In addition, $[Ca^{2+}]_i$ homeostasis can be restored by supplementing ATP directly to the cytosol of myocytes (Fig. 3e, f). These findings are corroborated by our computational modeling where replacing CKmito activity improved ATP levels in the

**Fig. 6 mtCaMKII activates TCA cycle dehydrogenases and augments resting NADH. a** Pyruvate dehydrogenase (PDH) activity in isolated mitochondria from WT ($n = 7$) and mtCaMKII ($n = 9$) hearts. **b** TCA cycle enzyme activities in permeabilized mitochondria from WT ($n = 7$) and mtCaMKII ($n = 9$) hearts, CS: citrate synthase, IDH: isocitrate dehydrogenase, α-KGDH: α-ketoglutarate dehydrogenase, SL: succinate-CoA ligase, SDH: succinate dehydrogenase, MDH: malate dehydrogenase. **c** Representative NADH imaging of isolated ventricular myocytes before pacing (calibration bar = 50 μm). **d** Quantification of NADH autofluorescence in isolated ventricular myocytes before and after pacing from WT ($n = 29$ myocytes from two hearts) and mtCaMKII ($n = 52$ myocytes from three hearts) hearts. Maximum (100%) NADH was measured in the presence of NaCN, and minimum (0%) in the presence of FCCP. Cells from the same genotype were paired in comparisons of before and after pacing. **e** Quantification of NAD$^+$ and NADH in hearts from WT ($n = 4$) and mtCaMKII ($n = 4$) mice. **f** Western blot and **g** summary data for acetylated proteins normalized to VDAC1 in isolated mitochondria from WT ($n = 4$) and mtCaMKII ($n = 4$) hearts. Data are represented as mean ± SEM, significance was determined using a two-tailed Student's *t* test. ****$P < 0.0001$, **$P < 0.01$, *$P < 0.05$. Source data, including exact *p* values, are provided as a Source data file.

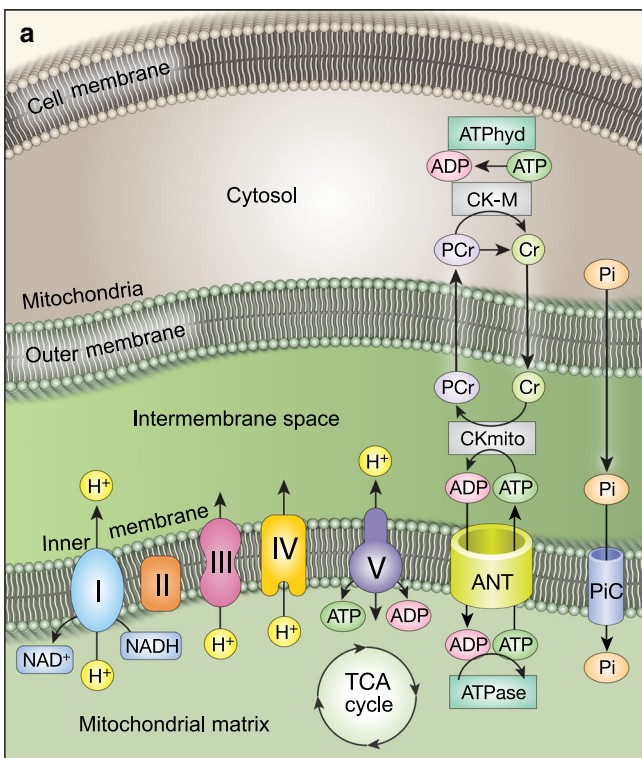

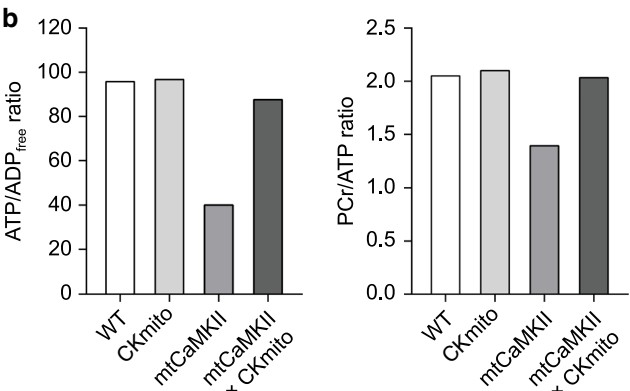

**Fig. 7 Computational modeling of energetics. a** Schematic image of key parameters used in a three compartment model of CaMKII, creatine kinase (CK), and mitochondrial energetics. **b** Computer stimulation results were compared in WT, CKmito, mtCaMKII, and mtCaMKII x CKmito interbred conditions. The ratios of PCr/ATP and ATP/ADP were calculated at the end of 5 min cardiac cycle (simulated using pulsatile functions to oscillate ATP hydrolysis and cytosolic calcium).

mtCaMKII model (Fig. 7 and Supplementary Fig. 6). These data suggest that the lack of adequate energy can directly lead to a stable pattern of ventricular dilation and contractile dysfunction, and restoring myocardial energy supply significantly improves cardiac function, despite persistence of excessive mitochondrial CaMKII activity.

Our computational modeling confirmed that ATP deficiency in mtCaMKII hearts is also partly a consequence of loss of complex I activity, and persists despite hyperactivation of the TCA cycle, and increased NADH. Our phosphoproteomic studies identified a site for PDH phosphorylation that is different from the well-known inhibitory sites regulated by mitochondrial calcium[71]. We observed enhanced PDH activity in the mtCaMKII hearts, but further investigation will be required to know if this phosphorylation site contributes to PDH activation under conditions of excessive CaMKII activity. Diminished complex I content is noted in ischemic and genetic forms of cardiomyopathy[68,70]. Our phosphoproteomic studies also identified NDUFB11, a complex I subunit essential for complex assembly and stability[72,73], as a CaMKII phosphorylation target, raising the possibility that excessive mitochondrial CaMKII activity has adverse consequences for complex I assembly. Other models of complex I deficiency show reduced sirtuin activity due to loss of the cofactor NAD$^+$, and it is possible that increased complex II activity was protective against excessive loss of NAD+ in mtCaMKII hearts compared to other models marked by loss of complex I. Loss of sirtuin mediated deacetylase activity results in excessive acetylation of mitochondrial proteins, mPTP opening, increased ROS, and myocyte death[62]. We did not find a significant deficiency of NAD$^+$ in mtCaMKII hearts, and the increases in acetylation in the mtCaMKII mitochondria were evidently insufficient to trigger loss of ΔΨ$_{mito}$ and cell death. Further studies will be needed to understand how reduced ATP, elevated [Ca$^{2+}$]$_i$, and, perhaps, other factors related to excessive mitochondrial CaMKII activity lead to left ventricular dilation. It does seem clear that the increased diastolic [Ca$^{2+}$]$_i$ observed in the mtCaMKII cardiomyocytes was insufficient to substantially activate Ca$^{2+}$ sensitive hypertrophic signaling pathways. Our current findings suggest mitochondrial CaMKII can contribute to pathological injury by remodeling metabolism.

Our results support a view that organelle resident and subcellular specific actions of CaMKII can selectively drive distinct cardiomyopathy phenotypes. Nuclear CaMKII overexpression causes mild myocardial hypertrophy[12], and non-targeted CaMKII overexpression leads to massive hypertrophy, myocardial death, fibrosis, chamber dilation, heart failure, deranged Ca$^{2+}$ homeostasis, and early sudden death[13,52]. Here, we show that targeted mitochondrial CaMKII overexpression causes a surprisingly pure dilated cardiomyopathy linked to adverse metabolic remodeling, and ATP deficiency. Unexpectedly, chronic actions of excessive

mtCaMKII do not cause increased myocardial or sudden death. We have not yet identified a mechanism for CaMKII translocation to mitochondria, but note, similar to the case of CaMKII, that multiple mitochondrial proteins lack canonical TOM/TIM guide sequences[74]. We assume mitochondrial CaMKII is activated by $Ca^{2+}$/calmodulin, ROS and $O$-GlcNAcylation, upstream CaMKII activating signals that are present in mitochondria[75,76]. The sequence of compartment specific increases in CaMKII activity in heart following injury are unknown, but our findings suggest that cellular stress pathways, activated by MI, likely contribute to increased mitochondrial CaMKII activity. The potential for CaMKII to operate in various cellular compartments suggests that CaMKII functions as a highly versatile agent, promoting hypertrophy and cardiac chamber dilation, depending on its subcellular location.

Failing human myocardium shows elevated diastolic $[Ca^{2+}]_i$ and increased cytosolic CaMKII activity, as evidenced by ryanodine receptor hyperphosphorylation and sarcoplasmic reticulum $Ca^{2+}$ leak[77]. Acute CaMKII inhibition is sufficient to reduce $Ca^{2+}$ leak, lower diastolic $[Ca^{2+}]_i$, and improve mechanical function in failing human myocardial strips[77]. Our studies extend the connections between CaMKII and cellular $Ca^{2+}$ homeostasis by demonstrating that loss of cytoplasmic $[Ca^{2+}]_i$ homeostasis and left ventricular dilation can arise from defective myocardial energetics, independent of increased cytoplasmic CaMKII activity because cardiac chamber dilation persisted in mtCaMKII mice interbred with AC3-I transgenic mice with extramitochondrial CaMKII inhibition (Fig. 2c). Myocardium is highly dependent on a moment to moment balance of ATP production and utilization related, in part, to the high-energy demands of continuous and physiologically changing work[78]. Achieving physiological $[Ca^{2+}]_i$ homeostasis is a major energetic task for myocardium[79], and elevation of diastolic cytoplasmic $[Ca^{2+}]$ is a common finding in animal heart failure models[80] and in failing human myocardium[81], conditions where energy supply is insufficient to satisfy demand. The mtCaMKII hearts show a severe loss of ATP and reduction in CKmito and complex I, consistent with a scenario where energy stores are inadequate to support normal function. ATP flux is purposed to specific subcellular compartments, and mitochondrial ATP is primarily responsible for fueling SERCA2a activity that is required to stabilize diastolic cytoplasmic $[Ca^{2+}]$ after each heartbeat[82,83]. Myofilament activation and cardiac contraction follow a regenerative release of $Ca^{2+}$ to the cytoplasm from the intracellular sarcoplasmic reticulum store, while relaxation and diastole are a consequence of sequestration of cytoplasmic $Ca^{2+}$ into the sarcoplasmic reticulum by SERCA2a. We found that decreased ATP in mtCaMKII hearts was associated with elevation of diastolic cytoplasmic $[Ca^{2+}]$, and that elevated $[Ca^{2+}]$ could be repaired in vitro by exogenous ATP and in vivo by replacement of CKmito. We interpret these findings to provide strong support for the hypothesis that mtCaMKII dilated cardiomyopathy arises from ATP deficiency that is exacerbated by CKmito deficiency. We speculate that sustained hyperactivation of mitochondrial CaMKII could contribute to metabolic defects and elevated diastolic $[Ca^{2+}]$ in human heart failure. In addition, others have shown that increases in myocardial ADP and intracellular $[Ca^{2+}]$ can contribute to diastolic dysfunction[84,85]. While we did not directly quantify diastolic function, this may also be contributing to the phenotype of the mtCaMKII mice.

The core components of adverse cardiac remodeling are myocardial hypertrophy and dilation[26,27]. However, the biological connections between hypertrophy and dilation are uncertain. One paradigm is that myocardial hypertrophy is a compensatory reaction to pathological stress, but sustained and excessive hypertrophy can progress to chamber dilation over time.

Although the interrelationship between hypertrophy and dilation is not completely understood, dilation may be favored by myocyte death interposed upon hypertrophied myocardium that eventually impairs myocardial ability to sustain elevated ventricular wall stress. However, the rate of progression to left ventricular dilation in patients with left ventricular hypertrophy detectable by sonographic imaging is low[86], suggesting that the processes of hypertrophy and dilation are not tightly coupled, nor necessarily driven by the same biological mechanisms. Our findings regarding mtCaMKII cardiomyopathy indicate that myocardial dilation can arise as a sustained response to myocardial energy deprivation in the absence of myocardial hypertrophy, or death. The cardiac dilation in mtCaMKII hearts is potentially reversible, because it does not involve increases in myocardial death, or fibrosis, and can be rescued by replacement of CKmito. Furthermore, $[Ca^{2+}]_i$ homeostasis can be repaired in ventricular myocytes isolated from mtCaMKII hearts by addition of ATP. Thus, it is possible that CKmito replacement, or mitochondrial CaMKII inhibition could prevent or reverse some acquired forms of dilated cardiomyopathy, a life-threatening disease phenotype that has been largely considered irreversible.

Many studies have provided evidence affirming a role of excessive CaMKII in myocardial disease. However, relatively less is known about the physiological roles of CaMKII in heart. Surprisingly, CaMKII activity appears dispensable for basal cardiac function[15,22], although complete ablation of CaMKIIγ and CaMKIIδ in myocardium causes mild hypertrophy due to loss of a tonic negative regulatory action on calcineurin signaling[87]. We interpret the available evidence to suggest that CaMKII promotes fight or flight responses that increase myocardial performance during extreme physiological stress. Many of these responses are linked to CaMKII catalyzed phosphorylation of $Ca^{2+}$ homeostatic proteins that enhance the excitation–contraction coupling mechanism. For example, CaMKII may augment the treppe effect in ventricular myocardium[88], and enhance the dynamic range of heart rate acceleration in cardiac pacemaker cells[89] by catalyzing ryanodine receptor phosphorylation. Our findings showing enhanced TCA cycle activity indicate that mitochondrial CaMKII can increase delivery of NADH to the electron transport chain. While we acknowledge that chronic overexpression of CaMKII creates an artificial condition, and although we currently lack tools to tightly control mitochondrial CaMKII expression or activity on a moment to moment timescale, we speculate that acute, physiological activation of mitochondrial CaMKII could be advantageous by increasing ATP flux from oxidative phosphorylation by enhanced NADH production through the TCA cycle. However, mtCaMKII mice, burdened by chronic, pathological elevation in mitochondrial CaMKII, exhibit loss of CKmito and complex I that prevent this metabolic benefit, and instead contribute to metabolic insufficiency, elevated diastolic cytoplasmic $[Ca^{2+}]$, and severe dilated cardiomyopathy.

## Methods

**Animal models**. All the experiments were carried out in accordance with the guidelines and approval of Institutional Animal Care and Use Committees at the University of Iowa and Johns Hopkins University (PHS Animal Welfare Assurance, A3021-01 (Univ. Iowa), A3272-01 (JHU)). Mice were housed in a facility with 14 h light/10 h dark cycle at $23 \pm 3\,°C$ and 30–70% humidity. Mice used in these studies were a mixture of male and female animals 7–20 weeks of age unless otherwise noted in the figure legend.

**Myocardial infarction**. Mice are anesthetized with 1.5–2% isoflurane and given a pre-emptive dose of buprenorphine (0.03–0.07 mg/kg). Mice are then intubated and ventilated (100% $O_2$, 200 μl TV, 120 bpm) using a Harvard Apparatus Mini-Vent. When a toe pinch confirms proper depth of anesthesia, mice are given a single dose of succinylcholine (2 mg/kg). Body temperature is maintained at 37 °C using a rectal thermometer and infrared heating lamp. A left thoracotomy and pericardiotomy is performed, and the left main coronary artery is completely

occluded by tying a knot with 7-0 prolene. After verification that MI has occurred (blanching of the tissue distal to the suture), the ribs and skin are closed with 5-0 silk. Once the chest is closed, the mouse remains under anesthesia until succinylcholine is completely metabolized. Isoflurane is turned off, the mouse is allowed to regain respiration on its own, and then extubated. When performing a sham surgery, mice undergo the same procedure, but the ligation of the artery is excluded. A second dose of buprenorphine (0.06–0.075 mg/kg) is given before returning mice to animal facility. The surgeon was blinded to the genetic identity of the mice for all studies.

**mtCaMKII mice.** The mouse CaMKII$\delta_C$ cDNA was fused with an N-terminal flag epitope tag and the N-terminal 28 amino acids of cox8a for mitochondrial targeting. The resulting construct was cloned into the pBS-$\alpha$MHC-script-hGH vector for myocardial expression. Pronuclear injections of linearized DNA (digested with NotI) were performed in the University of Iowa Transgenic Mouse Core Facility and embryos implanted into pseudo-pregnant females to generate C57Bl6/J F1 mice. Insertion of the transgene into the mouse genome was confirmed by PCR analysis using the forward primer, 5′-GCA GTC AGA AGA GAC GCG-3′, and reverse primer, 5′-GAA TCC CAA CAA CTC GGG AGG C-3′, producing a product of 500 bases.

**CKmito mice.** The CKmito transgenic mice were generated using aHMC-Tet promoter[90] fused with CKmito cDNA and human growth hormone polyA. The resulting TetCKmito mice were bred with aMHC-Tet-off mice to generate compound heterozygous mice as the CKmito transgenic mice[90,91]. Mouse genotypes were determined using real-time PCR with specific probes by a commercial vendor (Transnetyx, Cordova, TN). The CKmito transgenic mice express CKmito specifically in cardiomyocytes in the absence of doxycycline treatment.

**Echocardiography.** Transthoracic echocardiography was performed in unsedated mice using a Vevo 2100 (VisualSonics Inc) system, equipped with a 30–40 MHz linear array transducer. The left ventricle end-diastolic and end-systolic ventricular volumes (EDV, ESV), and the percent ejection fraction (EF) were estimated using the Simpson's method from the apical two chamber view of the heart[92,93]. Measurement of EDV, ESV were manually traced according the American Society of Echocardiography[94]. The EF was automatically calculated by the system building software (Vevo Lab 3.1.1) using the following formula:

$$EF\% = \frac{EDV - ESV}{100} * 100$$

The M-mode echocardiogram was acquired from the parasternal short axes view of the left ventricle (LV) at the mid-papillary muscles level and at sweep speed of 200 mm/s. From this view left ventricle mass (LV mass) is calculated from interventricular septal thickness at end of diastole (IVSD), LV chamber diameter at end of diastole (LVEDD), posterior wall thickness at end of diastole (PWTED) measurements using the following formula where 1.055 is the specific gravity of the myocardium[95]:

$$LV\,Mass\,(mg) = 1.055*\left((LSVD + LVEDD + PWTED)*3 - (LVEDD)^{*}3\right)$$

The echocardiographer was blinded to the genetic identity of the mice for all studies.

**In vivo murine cardiac MRI/MRS.** Anesthesia was induced under 2% isofluorane and maintained under 1% isofluorane and the in vivo studies carried out on a Bruker Biospec MRS/MRI spectrometer equipped with a 4.7 T/40 cm Oxford magnet and a 12 cm (i.d.) actively shielded BGA-12$^{TM}$ gradient set, as previously described[51,55]. For MRI for ventricular size and function, complete set of high temporal and spatial resolution multi-slice cine images were acquired and analyzed as previously reported[50,96,97]. LV EF was calculated from the difference in end-diastolic and end-systolic cavity volumes divided by the end-diastolic volume. For $^{31}$P MRS, after shimming, a one-dimensional $^{31}$P chemical shift imaging (1D-CSI) sequence, using adiabatic excitation pulses to obtain uniform flip angle across the field of view, was acquired as previously described[51]. The PCr and [β-P]ATP peaks in $^{31}$P MR localized spectra were quantified by integrating peak areas[50,96,97]. Voxel shifting, a processing method to shift the slice boundaries fractionally, permits the redefinition of the slice boundaries after data collection to align them with the LV border and minimize contamination from the chest wall, using high-resolution $^{1}$H images as a guide[98,99]. Concentrations of PCr and ATP as well as the ratio of PCr/ATP were measured as previously described[51]. The MRI studies were performed without knowledge of the genetic identities of the mice.

**Western blot.** Heart lysates were prepared in RIPA buffer containing protease and phosphatase inhibitors. Heart lysate or cytoplasm and mitochondria fractions were loaded into 4–12% Bis-Tris NuPAGE gels (ThermoFisher) or 4–15% Mini Protean gels (Bio-Rad). Proteins were transferred to nitrocellulose membrane using Turboblotter (Bio-Rad). Primary antibodies [CaMKII (Abcam), 1:1000; phospho-CaMKII (Thermo Scientific), 1:1000; VDAC1 (Abcam), 1:2000; FLAG (Rockland), 1:2000; GAPDH (Cell Signaling), 1:10,000; CKmito (Sigma), 1:6000; CoxIV (Cell

Signaling), 1:1000; CK-M (Sigma), 1:10,000; α-actinin (Sigma), 1:1000; OxPhos (Abcam), 1:500; AcLys (Cell Signaling), 1:1000; HA (Sigma), 1:5000; SERCA2a (Badrilla), 1:5000; PDH (Abcam), 1:1000] were incubated with the membrane overnight at 4 °C. Secondary antibodies were incubated with the membrane at room temperature for 1 h. Blots were imaged using an Odyssey Fc Imager (Licor). Bands were quantified using Image Studio Software (Licor), coomassie-stained membranes were quantified using Image J software.

**Mitochondrial isolation.** Mitochondria were isolated using homogenization method on ice as previously described[100]. In brief, the heart was minced and homogenized in ice-cold isolation buffer (75 mM sucrose, 225 mM mannitol, 1 mM EGTA, and 0.2% fatty acid free BSA, pH 7.4) using a Potter-Elvehjem glass homogenizer. The homogenates were centrifuged for 10 min at $500 \times g$. The resulting supernatants were centrifuged for 10 min at $10,000 \times g$. The mitochondrial pellets were washed twice in isolation buffer and centrifuged at $7700 \times g$ for 6 min each. Mitochondrial pellets were suspended in a small amount of isolation buffer to ~5 mg/ml. The protein concentration was determined using the BCA Assay kit (ThermoFisher Scientific).

**Mitochondrial ATP measurements.** Mitochondrial ATP content was measured using ATP Bioluminescence Assay Kit CLS II, Roche Life Science). Isolated mitochondria (15 μg) were suspended in KCl-based buffer at 37 °C. The ATP production was calculated from a standard curve for a series of ATP concentrations.

**Mitochondrial Ca$^{2+}$ content.** Total mitochondrial Ca$^{2+}$ content was measured in isolated mitochondria with the Calcium Assay Kit (Cayman Chemical). Total Ca$^{2+}$ was normalized to protein concentration.

**Mitochondrial ROS production.** Amplex UltraRed, a H$_2$O$_2$ sensitive fluorescent probe, was used to monitor ROS production in isolated mitochondria. Hydrogen peroxide was used to establish the standard curve as previously described[101]. Briefly, 15 μg of isolated mouse heart mitochondria was suspended in 0.2 ml KCl-based buffer solution (137 mM KCl, 0.25 mM EGTA, 2 mM MgCl$_2$, 2 mM KH$_2$PO$_4$, 20 mM HEPES, 5 mM NaCl, pH 7.2) with 4 U/ml horseradish peroxidase, 40 U/ml superoxide dismutase, 10 μM Amplex UltraRed in a 96-well plate, followed by sequential additions of substrate 5 mM pyruvate/malate, complex I inhibitor 0.5 μM rotenone, and complex III inhibitor 1 μg/ml anitimycin A. Amplex UltraRed fluorescent signal was read by a microplate reader (BioTek) at 535-nm excitation and 595-nm emission.

**Mitochondrial membrane potential.** Mitochondrial membrane potential ($\Delta\Psi_{mito}$) was monitored by the ratiometric dye tetramethylrhodamine methyl ester (TMRM) (546/573-nm excitation and 590-nm emission). Isolated mitochondria (25 μg) were suspended in 0.2 ml KCl-based buffer solution (137 mM KCl, 0.25 mM EGTA, 2 mM MgCl$_2$, 2 mM KH$_2$PO$_4$, 20 mM HEPES, 5 mM NaCl, pH 7.2) in a 96-well plate and read by using a BioTek Microplate reader. Glutamate/malate (5 mM/5 mM) were used as substrates to energize mitochondria (state 2) followed by addition of 1 mM ADP (state 3).

**Blue native gel and complex I activity.** Mitochondrial isolates from heart tissue lysates were prepared according to the NativePAGE Sample prep kit (ThermoFisher) (8 g/g: digitonin/protein). The mitochondrial resuspensions were run on 3–12% Bis-Tris Native Page gels (ThermoFisher). Gels were run using the NativePAGE Running Buffer kit (ThermoFisher). Gels were either stained using the Coomassie R-250 Staining kit (ThermoFisher) or used for in-gel activity. For this, the gel was equilibrated in 5 mM Tris HCl, pH 7.4 for 10 min and then developed in complex I activity solution (5 mM Tris pH 7.4, 2.5 mg/ml NBT, 0.1 mg/ml NADH). The reaction was terminated by soaking the gel in 10% acetic acid. Gels were imaged using an Epson Perfection V800 Photo scanner. For the microplate assay, Complex I activity was measured using the Complex I Enzyme Activity Microplate Assay Kit (Abcam ab109721) as specified by the manufacturer's instructions. Briefly, 100 μg mitochondrial pellets were resuspended in 25 μL protein extraction buffer and incubated on ice for 30 min. Samples were then centrifuged at $16,000 \times g$ for 20 min at 4 °C. Supernatant protein concentration was determined via BCA Assay kit (ThermoFisher Scientific). Twenty micrograms of protein were combined with Incubation Solution (Abcam) to a total of 1 mL. Each sample was loaded in triplicate (200 μL/well) and incubated for 3 h at room temperature. Complex I activity was determined by following change in 450 nm absorbance per minute (mOD/min) for 30 min following the addition of Assay Solution (Abcam).

**Mitochondrial respiration/oxygen consumption.** Mitochondrial oxygen consumption rate was assayed using a high-throughput automated 96-well extracellular flux analyzer (XF96; Seahorse Bioscience). Freshly isolated mitochondria (1–5 μg of mitochondrial protein) were suspended in the potassium-based buffer (137 mM KCl, 2 mM KH$_2$PO$_4$, 0.5 mM EGTA, 2.5 mM MgCl$_2$, and 20 mM HEPES at pH 7.2, and 0.2% fatty acid free BSA), transferred into a 96-well XF96 plate, and

centrifuged at $3000 \times g$ for 20 min at 4 °C to attach to the plate. The respiration was evaluated with substrates of complex I (5 mM pyruvate/malate), complex II (5 mM succinate), following addition of 1 mM ADP. Oxygen consumption was measured with one cycle of 0.5 min mix, 3 min measurement and another 0.5 min mix in each step after port injection[102].

**TCA cycle enzyme activities**. Pyruvate dehydrogenase (PDH) activity was measured with the Pyruvate Dehydrogenase Enzyme Activity Microplate Assay Kit (Abcam) using 20 μg of isolated mitochondria for each assay. Citrate synthase (CS) activity was measured using a MitoCheck Citrate Synthase Activity Assay Kit (Cayman Chemicals). The rest of TCA cycle enzymes activities were measured using colorimetric absorbance based kinetic assays as described[103]. Briefly, the isolated mitochondria were freeze/thawed three times to lyse the mitochondria. Mitochondrial lysate (15 μg) was then suspended in phosphate based solution (50 mM $KH_2PO_4$ with 1 mg/ml BSA, pH 7.2 in the first assay, 10 mM $KH_2PO_4$ in the second assay). The first assay used dichlorophenol Indophenol (DCPIP) reduction ($\lambda_{absorbance} = 600$ nm) to measure succinyl-CoA ligase (SL), succinate dehydrogenase (SDH), fumarase (FM), and malate dehydrogenase (MDH) by sequential addition of reagents 140 μM DCPIP, 100 μM Duroquinone, 0.8 mM phenazine methosulfate (PMS), 8 μM rotenone/100 μM succinyl-CoA, 0.2 mM KCN, 10 mM succinate, 10 mM malonate, 10 mM glutamate plus 800 μM $NAD^+$ and 1IU (AAT), 15 mM fumarate, and finally 10 mM malate. The second spectrophotometric assay measures α-ketoglutarate dehydrogenase (KGDH), aconitase (AC) and isocitrate dehydrogenase (IDH) by monitoring NADH/NADPH reduction in 800 μM $NAD^+$, 2 mM Dithiothreitol (DTT), 100 μM EDTA, 2 mM $CaCl_2$, 2 mM $MgCl_2$, 2 mM α-ketoglutarate, 100 μM thiamine pyrophosphate (TPP) and 0.1% Triton X100 by sequential additions of 0.5 mM CoASH, 0.8 mM $NADP^+$, 5 mM $MgCl_2$, 0.5 mM cis-aconic acid, and 15 mM isocitrate. Absorbance measurements were made using a BioTek Microplate reader (Synergy MX 96-well plate reader).

**Adult ventricular myocyte isolation**. Adult ventricular myocytes were isolated as previously described[104]. Briefly, mice (8–12 weeks, either gender) were anesthetized by Avertin injection. Hearts were rapidly excised and placed in ice-cold nominally $Ca^{2+}$ free HEPES-buffered Tyrode's solution. The aorta was cannulated, and the heart was perfused in a retrograde fashion with a nominally $Ca^{2+}$ free perfusate for 5 min at 37 °C. This was followed by a 15-min perfusion with collagenase-containing nominally $Ca^{2+}$ free solution. Final perfusion was with collagenase-containing low $Ca^{2+}$ (0.2 mM) solution. The LV and septum were cut away, coarsely minced and placed in a beaker containing low $Ca^{2+}$ solution with 1% (w/v) BSA at 37 °C. Myocytes were dispersed by gentle agitation, collected in serial aliquots and then maintained in standard saline solution containing 1.8 mM $Ca^{2+}$.

**Immunofluorescence imaging**. Isolated ventricular myocytes were fixed in 100% ethanol prior to incubation with antibodies Flag (Rockland), CoxIV (Cell Signaling). Stained cells were mounted on slides, and imaged on a laser-scanning confocal microscope (LSM 510 with Zen software, Carl Zeiss).

**Mitochondrial $Ca^{2+}$ uptake**. Mitochondrial $Ca^{2+}$ uptake was measured as previously described (Joiner, 2012) in saponin-permeablized ventricular myocytes. Assays were performed with freshly isolated myocytes in a 96-well assay plate in respiration buffer (100 mM KAsp, 20 mM KCl, 10 mM HEPES, 5 mM glutamate, 5 mM malate, and 5 mM succinate, pH 7.3) supplemented with 100 μM blebbistatin, 5 μM thapsigargin, 0.005% saponin, 5 μM CsA and 1 μM $Ca^{2+}$ green-5N (CaG5N, Thermo Scientific). CaG5N fluorescence was monitored (485-nm excitation, 535-nm emission) after adding $CaCl_2$ (50 μM free $Ca^{2+}$) at 3 min intervals at 30 °C. Cells from WT and mtCaMKII hearts were prepared on 3 separate days, and the mean of measurements read from 4 wells reported.

**Cytosolic $Ca^{2+}$ measurements**. Cytosolic $Ca^{2+}$ levels were recorded from Fura-2-loaded ventricular myocytes as previously described[82]. Briefly, single isolated ventricular myocytes were loaded with 4 μM Fura-2 acetoxymethyl (AM) for 30 min and then perfused with Tyrode's solution for 30 min to de-esterify the Fura-2 AM. After placement on a recording chamber, the cells were perfused in bath solution comprised (mM): 137 NaCl, 10 Hepes, 10 glucose, 1.8 $CaCl_2$, 0.5 $MgCl_2$, and 25 CsCl; pH was adjusted to 7.4 with NaOH, at $35 \pm 0.5$ °C. Whole-cell patch-clamp technique was used to dialyze cells. The pipette (intracellular) solution comprised of (mM): 120 CsCl, 10 Hepes, 20 TEA chloride, 1.0 $MgCl_2$, 0.05 $CaCl_2$, 10 glucose; the pH was adjusted to 7.2 with 1.0 N CsOH. Adenosine 5′-triphosphate disodium salt hydrate 5 mM was added to the pipette solution when needed. Myocytes were stimulated at 1, 3, and 5 Hz using a voltage protocol holding at −80 mV and stepping to 0 mV for 100 ms from a prepulse of 50 ms at −50 mV. The cytosolic $Ca^{2+}$ transients were measured from cells excited at wavelengths of 340 and 380 nm and imaged with a 510-nm long-pass filter. Data were collected and analyzed using NIS Elelements 4.00 software (Nikon).

**NADH measurements**. The autofluorescence of endogenous NADH was measured as described[83]. Briefly, ventricular myocytes were put into a recording

chamber on the stage of NiKon Eclipse Ti inverted microscope. NADH was excited at 350 nm (AT350/50X, Chroma) and fluorescence was recorded at 460 nm (ET460/50 m and T400LP, Chroma). We normalized NADH level with FCCP as 0%, Rotenone-induced NADH change as 100%. Data were collected and analyzed using NIS Elelements 4.00 software (Nikon).

**Transmission electron microscopy and mitochondrial scoring**. Hearts were fixed in 2.5% gluteraldehyde in 0.1 M Na cacodylate buffer overnight at 4 °C, washed $3 \times 20$ min in 0.1 M Na cacodylate buffer, pH 7.2, fixed in 4% $OsO_4$, washed in 0.1 M Na cacodylate buffer then $dH_2O$, 2.5% uranyl acetate, EtOH series to dehydrate then EtOH and Spurr's with final solution 100% Spurr's. Hearts were embedded in Spurr's at 60 °C for 24–48 h. Ultramicrotomy sections were cut at 90 nm and samples collected on 200 mesh for formvar grids for staining with uranyl and lead. Stained sections were examined with a Philips/FEI BioTwin CM120 Transmission Electron Microscope and digital images were collected with a Gatan Orius high-resolution cooled digital camera (2kx2k).

TEM images of mitochondria were scored by the following criteria: 0 = no detectable disruption in any mitochondria/field, 1 = cristae disrupted in one mitochondrion/field, 2 = disruption in >1 mitochondrion/field, 3 = ≥1 and <50% ruptured mitochondria/field and 4 = ≥50% mitochondria ruptured/field. At least 10 images were scored per sample by a blinded technician and the mean score reported.

**Quantitative PCR**. Total DNA was prepared using DNeasy Blood and Tissue Kit (Qiagen). Validated primers for Mitochondrial (CCCATTCCACTTCTGATTACC, ATGATAGTAGAGTTGAGTAGCG) and nuclear (GTACCCACCTGTCGTCC, GTCCACGAGACCAATGACTG) genes[105] were used for qPCR on CFX Connect™ Real-Time PCR Detection System (Bio-Rad) with CFX Manager 3.1 software (Bio-Rad). Mitochondrial to nuclear DNA ratios were quantified using the ΔΔCt method.

**TUNEL measurements**. Cryosections (10 μM) of ventricular tissue were fixed in 4% paraformaldehyde and stained with In-situ Cell Death Detection kit (Roche). Sections were imaged on a laser-scanning confocal microscope (LSM 510, Carl Zeiss). TUNEL positive and total nuclei were counted from five images per sample, and the averages reported.

**Fibrosis**. Mouse hearts were formalin fixed and paraffinized. Heart sections were cut along the coronal plane at a 60° for 30 min and deparaffinized through xylene, 100% ethanol, 95% ethanol and then water. The slides were then stained using a Masson's Trichrome Aniline Blue Stain Kit (Newcomer Supply, 9179B) and imaged at ×20 with an Aperio Scanscope CS. The images were then analyzed with Aperio ImageScope, using a lower intensity threshold of 150 and a hue value of 0.66. Six left ventricular free wall images were analyzed per sample. The positivity percentages of the six images were then averaged and the average positivity values were reported.

**Liquid chromatography and mass spectrometry**. Frozen extracted mitochondria were lysed by sonication in 300 μl of lysis buffer (8 M urea, 50 mM ammonium bicarbonate, 1X protease inhibitors (Complete mini EDTA-free mixture [Roche Applied Science], 1X phosphatase inhibitor mixture [PhosSTOP, Roche Applied Science]). After centrifugation ($20,000 \times g$ for 10 min at 4 °C), the protein concentration of the supernatant was measured using the Bradford assay (Bio-Rad) and ~150 μg of proteins for each condition were subjected to digestion. Protein reduction and alkylation were performed using a final concentration of 2 mM dithiothreitol and 4 mM iodoacetamide, respectively. Proteins were first digested for 4 h at 37 °C with Lys-C (enzyme/substrate ratio 1:100). The second digestion was performed overnight at 37 °C with trypsin (enzyme/substrate ratio 1:100) in 2 M Urea. The resulting peptides were chemically labeled using stable isotope dimethyl labeling as described previously[106]. After protein digestion the mtCaMKII mitochondria were labeled as "Intermediate", while the WT were labeled with "Light". Sample were mixed in a 1:1 ratio and ~300 μg of the peptide mixtures were subjected to phosphopeptide enrichment using $Ti^{4+}$-IMAC material as described previously[107]. Briefly, the mixtures of labeled samples were dried to completion and reconstituted in 80% ACN, 6% trifluoroacetic acid (TFA), and loaded onto the $Ti^{4+}$-IMAC columns. After washing with 50% ACN, 0.5% and 0.1% TFA, and 200 mM NaCl consecutively, the phosphopeptides were eluted first with 10% ammonia and then with 2% FA and 80% ACN, and were dried to completion in a vacuum centrifuge. After reconstitution in 10% FA, 5% dimethyl sulfoxide, the peptides were analyzed using nano-flow reverse phase liquid chromatography on a Proxeon Easy-nLC 1000 (Thermo Scientific) coupled to an Orbitrap Elite (Thermo, San Jose, CA). Peptides were separated on an in-house made 50 cm column, 75 μm inner diameter packed with 1.8 μm C18 resin (Agilent Zorbax SB-C18) at a constant temperature of 40 °C, connected to the mass spectrometer through a nanoelectrospray ion source. The injected peptides were first trapped with a double fritted trapping column (Dr Maisch Reprosil C18, 3 μm, 2 cm × 100 μm) at a pressure of 800 bar with 100% solvent A (0.1% formic acid in water) before being chromatographically separated by a linear gradient of buffer B (0.1% formic acid in acetonitrile) from 7% up to 30% in 170 min at a flow rate of 150 nl/min. Nanospray

was achieved with an in-house pulled and gold coated fused silica capillary (360-μm outer diameter, 20-μm inner diameter, 10-μm tip inner diameter) and an applied voltage of 1.7 kV. Full-scan MS spectra (from $m/z$ 350 to 1500) were acquired in the Orbitrap with a resolution of 30,000. Up to ten most intense ions above the threshold of 500 counts were selected for fragmentation. For the fragmentation a decision tree method was used as described previously[108]. The mass spectrometry proteomics data have been deposited to the ProteomeXchange Consortium via the PRIDE[109] partner repository with the dataset identifier PXD004631.

**LC-MS data analysis**. For the raw data files recorded by the mass spectrometer, peak lists were generated using Proteome Discoverer (version 1.3, Thermo Scientific, Bremen, Germany) using a standardized workflow. Peak list was searched against a Swiss-Prot database (version 2.3.02, taxonomy Mus musculus, 32402 protein entries) supplemented with frequently observed contaminants, using Mascot (version 2.3.02 Matrix Science, London, UK). The database search was performed by using the following parameters: a mass tolerance of 50 ppm for the precursor masses and ±0.6 Da for CID/ETD fragment ions. Enzyme specificity was set to Trypsin with two missed cleavages allowed. Carbarmidomethylation of cysteines was set as fixed modification, oxidation of methionine, dimethyl labeling (L, I) of lysine residues and N termini, and phosphorylation (S, T, Y) were used as variable modifications. Percolator was used to filter the PSMs for <1% false discovery-rate. Phosphorylation sites were localized by applying phosphoRS (pRS) (v2.0)[110]. Double dimethyl labeling was used as quantification method[111], with a mass precision of 2 ppm for consecutive precursor mass scans. A retention time tolerance of 0.5 min was used to account for the potential retention time shifts due to deuterium. To further filter for high-quality data, we used the following parameters: high-confidence peptide-spectrum matches, minimal Mascot score of 20, minimal peptide length of 6 only, and unique and the search rank 1 peptide. The phosphopeptides that showed an on/off situation in the mtCaMKII or WT were manually quantified by giving them an arbitrary value of 100 or 0.01 for extreme up- or downregulation, which corresponds to the maximum allowed fold change in the used Proteome Discoverer settings.

**Modeling and simulation**. A three compartment computational model was constructed to include mitochondrial matrix, intermembrane space (IMS), and cytosol compartments. The compartments were separated by inner mitochondrial membrane (IMM), and outer mitochondrial membrane (OMM). The computational model consists of 48 nonlinear ordinary differential equations. The mitochondrial processes include oxidative phosphorylation, TCA cycles and transporters across the mitochondrial inner membrane[63,64]. The mitochondrial creatine kinase in the intermembrane space (IMS) and in the cytosol were modeled as random sequential bi-bi enzyme reactions and creatine, creatine phosphate, ATP and ADP were diffused between IMS and cytosol[112]. The equations were solved by ODE solver ode15s in MATLAB 2018b (The MathWorks, Natick, MA). The computational model of mitochondrial and cellular CaMKII and creatine kinase system was constructed to simulate overexpressed mitochondrial CaMKII condition and mitochondrial creatine kinase (CK). The mitochondrial CaMKII overexpressing (mtCaMKII) condition was simulated by decreasing complex I activity to 45% of control, decreasing mitochondrial CK to 80% of control, and increasing complex II (130%) and TCA cycle enzymes activities (isocitrate dehydrogenase (IDH) 120%, α-ketoglutarate dehydrogenase (αKGDH) 120%, fumarate hydratase (FH) 120%, malate dehydrogenase (MDH) 20% more) according to the experimental measurements of these enzyme activities. Mitochondrial CK overexpression was simulated by increasing mitochondrial CK concentration to 120% of control. ATP hydrolysis and cytosolic calcium transient during cardiac systole was simulated using a pulsatile function. A detailed description of the mathematical equations used is available in the Supplementary Methods (see Model Validation).

**Statistical analysis**. Statistical analyses were performed using Graph Pad Prism 7 software. Sample size and information about statistical tests are reported in the figure legends. Data are presented as mean ± SEM. Pairwise comparisons were performed using a two-tailed $t$ test. For experiments with more than two groups, data were analyzed by one-way ANOVA followed by Tukey's post hoc multiple comparisons test.

**Reporting summary**. Further information on research design is available in the Nature Research Reporting Summary linked to this article.

## Data availability

The authors declare that the data supporting the findings of this study are available within the paper and its supplementary information files. Source data are provided with this paper. The mass spectrometry proteomics data are available via the ProteomeXchange Consortium via the PRIDE[109] partner repository with the dataset identifier PXD004631. The data that support the findings of this study and unique materials are available from the corresponding authors upon reasonable request. Source data are provided with this paper.

## Code availability

Simulation codes are available at the URL: https://gitlab.com/MitoModel/mtCaMKII.git.

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

## Acknowledgements

This work was supported by the NIH (R35 HL140034 to M.E.A. and HL63030 and HL61912 to R.G.W.), an American Heart Association Collaborative Sciences Award (17CSA33610107 to M.E.A.), MOST (Taiwan) (MOST-107-2636-B-002-001 to A.W.), a grant from the Fondation Leducq for the Alliance for CaMKII Signaling (M.E.A., A.J.R.H.), and the Netherlands Organization for Scientific Research (NWO) through funding of the large-scale proteomics facility *Proteins@Work* (project 184.032.201) embedded in the Netherlands Proteomics Centre (E.C. and A.J.R.H.). We thank Jinying Yang for animal model maintenance; Marwan Mustafa, Gianna Bortoli, Djahida Bedja, Michelle Leppo, and the Cardiovascular Physiology and Surgery Core at Johns Hopkins School of Medicine for technical assistance; Shawn Roach and Teresa Ruggle for graphic design and figure preparation.

## Author contributions

Conceptualization: E.D.L. and M.E.A.; software: W.T. and A.W.; methodology: Y. Wang; validation: E.D.L.; formal analysis: E.D.L., A.W., A.G., J.M.G, N.R.W., K.R.M., and P.U.; investigation: E.D.L, J.M.G., N.R.W., Y.Wu, A.W., A.G., E.C., M.A.J., A.S., O.E.R.G, K.R.M., and P.U.; data curation: E.D.L.; writing—original draft: E.D.L. and M.E.A.; writing—review & editing: E.D.L., M.E.A, R.G.W., and A.W.; supervision: M.E.A., A.J.R.H., and R.G.W.; project administration: E.D.L.; funding acquisition: M.E.A., A.J.R.H., A.W., and R.G.W.

## Competing interests

The authors declare no competing interests.
