## [Peer Review File · Nature Communications]

Reviewers' comments:

Reviewer #1 (Remarks to the Author):

Summary:

The manuscript titled "Mitochondrial CaMKII causes metabolic reprogramming, energetic insufficiency, and dilated cardiomyopathy" investigates the mechanisms linking pathological stress to the altered myocardial metabolism observed in the failing heart. Cytosolic CaMKII is a key mediator of cardiac hypertrophy and has been implicated in dilated cardiomyopathy, but little is known about the specific mechanisms by which CaMKII may contribute to the dilation process. Previous work by this group and others demonstrated that CaMKII is present in cardiac mitochondria, and that inhibition of CaMKII in the mitochondria protects against mPTP opening, loss of $\Delta\psi$, and cell death in myocardial ischemia reperfusion injury, suggesting a critical role for CaMKII in modulating mitochondrial function after an acute pathological insult. Given these data and the association between poor mitochondrial function and heart failure/dilation, here the authors investigated how mitochondrial CaMKII activity specifically affects chronic changes in myocardial energetics and pathological remodeling post-MI.

Using genetic mouse models of mitochondrial CaMKII overexpression or inhibition, the authors present data showing that increased mitochondrial CaMKII activity persists 1 week post-MI, when hearts are becoming dilated, and that inhibition of mitochondrial CaMKII blocks MI-induced cardiac dilation and dysfunction. Furthermore, genetic overexpression of mitochondrial CaMKII alone is sufficient to cause spontaneous dilated cardiomyopathy, which is associated with compromised myocardial energetics that ultimately impairs cytosolic calcium handling that contributes to contractile dysfunction. Finally, the authors demonstrate that replenishment of mitochondrial creatine kinase in mice with mitochondrial CaMKII overexpression is sufficient to improve myocardial bioenergetics and partially rescue the dilated cardiomyopathy phenotype.

Major Concerns:

1. More data is needed to clearly show that mito-CaMKII phosphorylation is increased after injury. A cytosolic loading control and also showing cytosolic CaMKII phosphorylation would help bolster the claim that this is indeed mito-localized. Perhaps the inclusion of sub-fraction assays using protease degradation to show this is matrix localized would help this claim. Can CaMKII activity be examined in the mito localized fraction?

2. The results shown in Figure 1C are astounding. It is virtually impossible to see such a huge protective effect only 1 week after permanent LCA ligation where no reperfusion is present.

This model lends itself to see changes in LV remodeling/infarct expansion, but not so much to acute changes in structure or function. A couple mice in the mtCaMKIIN group appear to have almost no dysfunction 1wk post MI. Is the ligation/infarct size the same in these groups? The differences at 1 week are tough to believe. The authors may want to make sure all mice in the study received a similar infarct.

3. For all direct measurements of HW/BW used to demonstrate effects on hypertrophy (Fig. 1G, etc), please show raw data (HW and BW measurements separately), and/or HW/TL measurements. Given the dramatic cardiac phenotype in the mtCaMKII mice, is body weight changing between genotypes? Direct histological measurements of cardiomyocyte cross sectional area would also be informative here and help to strengthen claims about mtCaMKII's effect on hypertrophy.

4. Cytoplasmic marker is needed in Fig 2A.

5. The data interpretation in Figure 2 that the phenotype is mitoCamKII dependent is not clear to me. The inhibitor AC3-I is almost entirely localized to the cytosol yet it is significantly improving EF and ESV in the mtCaMKII mice. It appears that much of the phenotype is due to cytosolic activity especially since the inhibitor is probably not 100% effective.

6. The 2nd line of evidence is shown in Figure D but the authors do not show the percent of mtCAMKIIN that is in the cytosol vs. the mitochondria.

7. The data showing a difference in ATP and pCRr levels is at a time with massive cardiac pathology. It's difficult to attribute this as a direct result of mtCaMKII and not secondary to the severe phenotype.

8. The data regarding diastolic calcium correction with ATP is nice (Fig 3E,F). Can the authors provide the same data with different pacing for amplitude, rise-time and decay? It's a little difficult to understand that if there is insufficient ATP to manage diastolic calcium (likely an effect at SERCA, since it is the most ATP dependent) how the individual transient decay is not affected.

9. Fig 4A-D. How well do the CKmito and CK-M antibodies differentiate these two isoforms? The blots shown here are from total heart lysates, so it is difficult to assess if CK in the mitochondria

specifically is truly altered in mtCaMKII hearts, or whether the CKmito and CK-M antibodies might be picking up both myofibrillar and mitochondrial isoforms. Showing cytosolic vs. mitochondrial fractions would be helpful here or since there are not significant differences these data could be disregarded.

10. Is there a significant increase in CK protein in the CK-mito Tg mouse? Did the authors also examine mRNA? Is there a tag on the transgene that can be exploited?

11. Fig. 4. Is increased CKmito phosphorylation associated with any alterations in CKmito activity? And is loss or inhibition of CKmito alone sufficient to drive diminished mitochondrial energetics and contractile dysfunction/dilation seen in mtCaMKII mice? Is CKmito a direct phosphorylation target of mtCaMKII?

12. Fig. 4G. How does CKmito expression in mtCaMKII hearts recover both ATP content and PCr content, rather than just shifting the balance between the two? How does CKmito increase the total pool of high energy phosphate species in this context?

13. Fig 5. Does CKmito expression in the mtCaMKII mice rescue Complex I activity as measured in-gel or by OCR, in addition to rescuing complex I expression? What is the link with complex I expression? How does the CK-mito rescue have such a big effect on complex I formation?

14. How does the PDH activity increase in mtCaMKII hearts correlate with the increase in phosphor-PDH reported in the mass-spec data, since phosphor-PDH is inhibited? Is this the inhibitory site? Also, there was no change in mitochondrial calcium which is a strong physiological regulator of PDH activity.

Reviewer #2 (Remarks to the Author):

Luczak and colleagues found that mitochondrial CaMKII is activated upon myocardial infarction and that targeted inhibition of CaMKII in mitochondria can preserve cardiac function. They then generated a model of CaMKII overexpression in mitochondria and show a phenotype of ATP depletion and dilated cardiomyopathy which can be rescued by creatine kinase re-expression. This is an interesting manuscript describing effects of CaMKII on mitochondrial energy production in association with dilatation of the myocardium. Many findings are novel and advance the field.

Major comments that should be addressed:

1. Fig. 1A: To assess CaMKII activity in MI, additional measures other than pCaMKII should be used because pCaMKII as measure has been reported to be limited due to cross reactivity of most of the antibodies used in the field. Can the authors measure mitochondrial CaMKII phosphorylation targets as they do in Fig. 2G for mtCaMKII, e.g. MCU phosphorylation?
2. Moreover, looking more closely to the $\Delta C / \Delta B$ ratio in Fig. 1A, it looks alike a previous finding of Weinreuter et al in 2014 (EMBO Mol Med, Fig. 2A) that the ratio shifts towards ΔC . Can the authors quantify this in their own experiment again?
3. Can the authors show that hyperphosphorylation of the MCU upon MI is inhibited by mtCaMKIIIN to judge whether CaMKII is indeed inhibited in this model?
4. Fig. 1C: The authors show that mtCaMKIIIN mice are protected from cardiac dysfunction one week after myocardial infarction. But isn't that expected in light of the prior paper that these mice have less infarct size in the I/R model? Have the authors measured initial infarct size or troponin T levels in this experiment? This is important to assess the relative contribution of different CaMKII inhibition-dependent protective effects on cardiac function.
5. The authors refer to refs 15 and 21 to say that CaMKII is dispensable for cardiac function but in these studies only CaMKII delta was deleted and CaMKII activity was not completely abolished. They then correctly also refer to Kreusser et al. (ref 76) because in this paper no residual CaMKII activity was detectable because the two redundant isoforms CaMKII delta and gamma were knocked out simultaneously. The same mouse model without any CaMKII activity was also used by Weinreuter et al. in an extensive ischemia/reperfusion study. How do the authors discuss their own findings in light of the Weinreuter paper? In that paper cardiac function upon I/R was protected only at later stages most likely to a reduced pro-inflammatory burden. Can the authors measure also pro-inflammatory cytokines and leucocyte infiltration in their model as alternative causes of contractile dysfunction?
6. Loss of creatine as shown by the Neubauer lab does not cause exacerbation due to MI. How do the authors explain this in relation to the own findings?

Minor:

1. MW markers should be shown in the Western blot experiments

In summary, I recommend a major revision.

Reviewer #3 (Remarks to the Author):

The paper by Luczak et al. investigates the role of mitochondrial calcium/calmodulin-dependent protein kinase II (CaMKII) in cardiac myocytes and its role in regulating mitochondrial function during pathology. First, the authors observed that one week after myocardial infarction (MI), mitochondrial CaMKII expression and phosphorylation (activity) is increased. Genetic inhibition of mitochondrial CaMKII protects from left ventricular (LV) dilation and dysfunction after MI. Mice with overexpression of mitochondrial CaMKII in the heart have dilated cardiomyopathy with energetic deficit. This can be rescued when mitochondrial creatine kinase (mitCK) is overexpressed in parallel. The authors conclude that these insights provide two new strategies how to prevent LV dilation after MI, i.e., increasing mitoCK or inhibiting mitochondrial CaMKII.

The study is generally very well conducted and addresses a clinically important topic, i.e., heart failure after MI, one of the most frequent causes for cardiovascular and total deaths worldwide. Furthermore, gaining insights into mitochondrial dysfunction after ischemia/reperfusion may also be of broader interest since similar mechanisms may (or may not) be in place after stroke or ischemia of other regions of the body.

The findings of the paper are indeed novel. The manuscript is nicely written, and the Figures are clear and well organized. A particular strength of the study is the combination of extensive genetic work on in vivo mouse models with functional experiments on isolated mitochondria and cardiac myocytes. The findings would have an important impact on the field, since the control of mitochondrial function in particular, after MI and in the context of maladaptive remodeling, is not fully understood, and CaMKII is a key regulatory hub that receives widespread attention in the scientific community.

However, the study has also several shortcomings that limit my enthusiasm about the conclusions drawn. These are i) the mere use of transgenic techniques to overexpress and target certain proteins

in cardiac myocytes (which may be more artificial) rather than using knock-out technology, ii) the fact that no clear (molecular) mechanism of how CaMKII enters mitochondria upon stress and influences mitochondrial function under physiological or pathological situations is proposed, and iii) some of the conclusions drawn are not (yet) fully supported by the experiments. In the following, I will address these issues in more detail:

Major:

1) Figure 1: The localization of CaMKII in mitochondria is not very convincing yet. It is a technical challenge to isolate mitochondria without contamination by the ER/SR, based on a close interaction and tethering of these two organelles. Can you verify that the CaMKII expression and phosphorylation assessed in isolated mitochondria stems from a pure mitochondrial isolation? You show the presence of VDAC, verifying mitochondrial protein, but do not show the absence of SR-related proteins.

2) Another important information, also in view of your results on regulation of mitoCK and Krebs cycle enzymes by CaMKII (see below), is the actual localization of CaMKII within mitochondria (inner or outer mitochondrial membrane, matrix). In the introduction, you mention matrix localization. For this, further biochemical experiments with differential digestion protocols would be required to determine the actual localization of native CaMKII. The same applies to the transgenic overexpressed and targeted mitoCaMKII (Figure 2A). The immune-fluorescence colocalization of CaMKII with Cox IV (Figure 1E) alone hardly answers this, especially since CoxIV is in the inner mitochondrial membrane. If CaMKII resides indeed in the matrix, how would it be regulated from upstream, and would that be via signals stemming from inside or outside the mitochondria?

3) The transgenic overexpression of CaMKII in mitochondria is rather an artificial condition. This should be at least discussed, since in this project, no KO strategy is used, which would provide a somewhat cleaner information. This is of particular relevance because in some aspects, your previous results on reduction of infarct size by mitochondrial CaMKII inhibition differ from observations in CaMKII delta/gamma double KO mice (Weinreuter et al., EMBO Mol Med 2014), in which infarct size was unaffected by the KO (although the subsequent remodeling was ameliorated).

4) Figure 2I-J: The observation that overexpression of mitochondrial CaMKII does not affect MCU-mediated Ca uptake stands at odds with your previously proposed concept in which CaMKII actually does control MCU-mediated mitochondrial Ca uptake (Joiner, Nature). In this context, did you also re-test whether CaMKII-IN expression affects mitochondrial Ca uptake? And from a technical side, you may get a cleaner view on the actual mitochondrial Ca uptake if you perform the Ca retention assays on isolated mitochondria in the presence of cyclosporine A, since differential PTP activation (potentially controlled by CaMKII?) could blur your view on uptake by affecting Ca release via PTP. Therefore, I suggest to perform these experiments in the absence and presence of CsA.

5) Figure 2L: In the methods you mention sequential additions of pyruvate/malate, rotenone and antimycin A. Which of these conditions do you show in the figure? Can you show all? Did you also measure H₂O₂ emission in the presence of ADP to simulate more physiological (respiring) conditions? Mitochondrial H₂O₂ emission is controlled also by NADPH and the transhydrogenase (Nnt), which is not expressed in BL/6J mice. Do you know about the expression of the Nnt in your various mouse models? This may have an impact on the results of H₂O₂ emission and, if indeed most are in BL/6J background, should be at least critically discussed.

6) Figure 3: Why did you use Cs-based bath and pipette solutions in your experiments determining cytosolic Ca transients? Would it not be better to allow EC coupling as during physiological conditions? And since the in vivo phenotype is a systolic form of heart failure, I would be interested in the amplitudes and kinetics of cytosolic calcium transients as well; from the traces you show one could imagine that in the mitoCaMKII, they are somewhat reduced?

7) You mostly argue with reduced ATP, but you may consider that also an accumulation of ADP per se has a substantial impact on diastolic function (see Sequeira et al., J Physiol 2015), since ADP can induce diastolic dysfunction on the sarcomere level. Also, a change in the ATP/ADP ration and thereby, ΔG ATP may impact the rate of SERCA, and SERCA function may be reflected by the rates of calcium decay in your cell experiments. Therefore, please add the kinetics to these results.

8) Figure 5: Do you have any idea how complex I could be downregulated so strongly by mtCaMKII overexpression, and how changes of ATP could upregulate the expression of complex I (since CKmito expression partially improves complex I expression)?

9) Figure 6: Your results of NADH/NAD⁺ in resting cardiac myocytes make sense, since in light of the upregulation of TCA cycle enzymes and downregulation of complex I, one indeed expects a more reduced redox state of NADH/NAD⁺. However, I did not understand why NADH becomes more oxidized in response to pacing, which is even more a situation of electrons flowing from the Krebs cycle into the respiratory chain and thereby, should aggravate any alterations in redox state which may be related to the alterations in complex I and Krebs cycle enzymes, respectively. Furthermore, one can assume that during pacing, the actual redox state of NADH/NAD⁺ is similar between both groups, since the WT start more oxidized but become more reduced, while the mtCaMKII start more reduced but become oxidized. So what is the net steady-state redox state of NADH/NAD during pacing? Finally, if you collect also the redox state of FADH₂/FAD in these myocytes, you may get a good idea how your complex II upregulation affects the redox state of that pool (since FADH₂ fuels into complex II).

10) Figure 7: The ATP/ADP ratio is rather small, I would expect values of >100?

11) You mention that in mtCaMKII mice, CKmito is hyperphosphorylated. How specific is this result? Is there a known phosphorylation site of CK by CaMKII? Is this phosphorylation blunted in CaMKII KO mice? And is CKmito downregulated in HF (for instance, in your MI model)?

We would like to thank the reviewers for their time and constructive comments. The new manuscript is substantially improved by the responsive revisions, in many cases employing new experimental data. Here we address the critiques in a point-by-point fashion. Text responses and changes to the original manuscript are marked by yellow highlighting.

Reviewer #1 (Remarks to the Author):

Summary:

The manuscript titled “Mitochondrial CaMKII causes metabolic reprogramming, energetic insufficiency, and dilated cardiomyopathy” investigates the mechanisms linking pathological stress to the altered myocardial metabolism observed in the failing heart. Cytosolic CaMKII is a key mediator of cardiac hypertrophy and has been implicated in dilated cardiomyopathy, but little is known about the specific mechanisms by which CaMKII may contribute to the dilation process. Previous work by this group and others demonstrated that CaMKII is present in cardiac mitochondria, and that inhibition of CaMKII in the mitochondria protects against mPTP opening, loss of delta psi, and cell death in myocardial ischemia reperfusion injury, suggesting a critical role for CaMKII in modulating mitochondrial function after an acute pathological insult. Given these data and the association between poor mitochondrial function and heart failure/dilation, here the authors investigated how mitochondrial CaMKII activity specifically affects chronic changes in myocardial energetics and pathological remodeling post-MI.

Using genetic mouse models of mitochondrial CaMKII overexpression or inhibition, the authors present data showing that increased mitochondrial CaMKII activity persists 1 week post-MI, when hearts are becoming dilated, and that inhibition of mitochondrial CaMKII blocks MI-induced cardiac dilation and dysfunction. Furthermore, genetic overexpression of mitochondrial CaMKII alone is sufficient to cause spontaneous dilated cardiomyopathy, which is associated with compromised myocardial energetics that ultimately impairs cytosolic calcium handling that contributes to contractile dysfunction. Finally, the authors demonstrate that replenishment of mitochondrial creatine kinase in mice with mitochondrial CaMKII overexpression is sufficient to improve myocardial bioenergetics and partially rescue the dilated cardiomyopathy phenotype.

Thank you for your excellent summary, positive comments, and thorough review of our manuscript.

Major Concerns:

- 1. More data is needed to clearly show that mito-CamKII phosphorylation is increased after injury. A cytosolic loading control and also showing cytosolic CamKII phosphorylation would help bolster the claim that this is indeed mito-localized. Perhaps the inclusion of sub-fraction assays*

using protease degradation to show this is matrix localized would help this claim. Can CamKII activity be examined in the mito localized fraction?

In response to your comment, we repeated the 1 week post-MI studies. We now include a cytosolic, matrix and SR loading control in revised Fig 1A, and present the pCaMKII (autophosphorylation) immunoblots from the cytosolic fractions (Fig S1A). Additionally, we measured phosphorylated MCU (p-MCU), a mitochondrial CaMKII target (Nguyen et al. 2018), and show an increase in p-MCU 1 week after MI compared to sham hearts (revised Fig 1A and B; page 3, paragraph 2). We interpret these data to further support our original findings that increased mitochondrial CaMKII persists one week after MI.

2. The results shown in Figure 1C are astounding. It is virtually impossible to see such a huge protective effect only 1 week after permanent LCA ligation where no reperfusion is present. This model lends itself to see changes in LV remodeling/infarct expansion, but not so much to acute changes in structure or function. A couple mice in the mtCaMKIIN group appear to have almost no dysfunction 1wk post MI. Is the ligation/infarct size the same in these groups? The differences at 1 week are tough to believe. The authors may want to make sure all mice in the study received a similar infarct.

We agree that MI surgery has potential for variability. To better address this issue and to respond to your comments, we repeated a series of MI studies (n = 9-10/group) using the same highly experienced surgeon who performed the original studies. In all studies she was blinded to the genetic identity of the mice. In these new studies, we measured infarct size in mtCaMKIIN and WT littermate hearts 24 hours after coronary artery ligation to assess the initial size of the injury (revised Fig S1C and revised Supplemental Methods; Supplemental Information page 2). We saw no significant difference in the amount of infarcted myocardium. Additionally, we obtained echocardiographic measurements from a new cohort of mice (n=6-9/group) 1 week following coronary artery ligation. These new studies confirmed that the mtCaMKIIN expressing hearts were highly protected from myocardial dysfunction compared to WT littermate controls. In all cases the echocardiographer was blinded to the genetic identities of the mice. These new data were combined with the previous cohort and shown in revised Fig 1C.

3. For all direct measurements of HW/BW used to demonstrate effects on hypertrophy (Fig. 1G, etc), please show raw data (HW and BW measurements separately), and/or HW/TL measurements. Given the dramatic cardiac phenotype in the mtCaMKII mice, is body weight changing between genotypes? Direct histological measurements of cardiomyocyte cross sectional area would also be informative here and help to strengthen claims about mtCaMKII's effect on hypertrophy.

Based on your comments, we included the data for HW, BW, and HW/tibia length in revised Fig S2A. There is a trend toward increased HW in the mtCaMKII, as suggested in our original manuscript, and a similar trend was present in HW/TL between mtCaMKII mice and their WT

littermates. There was not a difference in BW between the WT and mtCaMKII mice. We measured cross-sectional area of cardiomyocytes in sections from mtCaMKII and WT littermate hearts, and found no significant difference in myocyte size between the two genotypes (n=4-5 hearts/group; n=1375-2733 myocytes/heart). These new data are included in revised Fig S2B-C (page 4, paragraph 3) and in the revised Supplemental Methods section (Supplemental Information, page 3). Overall, these new data confirm our original findings that mtCaMKII hearts are massively dilated, but with minimal hypertrophy.

4. Cytoplasmic marker is needed in Fig 2A.

We added a panel with GAPDH to revised Fig 2A.

5. The data interpretation in Figure 2 that the phenotype is mitoCamKII dependent is not clear to me. The inhibitor AC3-I is almost entirely localized to the cytosol yet it is significantly improving EF and ESV in the mtCaMKII mice. It appears that much of the phenotype is due to cytosolic activity especially since the inhibitor is probably not 100% effective.

We apologize for the lack of clarity around this point. Although CaMKII over-expression is primarily targeted to mitochondria in the mtCaMKII hearts, there is a minor amount of 'leak' to the cytosol (Fig 2A). We used AC3-I transgenic mice as a tool to determine the potential role of the cytosolic component of transgenically expressed (flag-tagged) CaMKII in the myocardial dilation and dysfunction seen in mtCaMKII mice; ideally for this purpose, the transgenically expressed AC3-I/eGFP fusion protein is excluded from the mitochondrial fraction (Fig 2B). The mtCaMKII x AC3-I interbred hearts (a model of mitochondrial CaMKII activation and extramitochondrial CaMKII inhibition) show a partial rescue of the reduced EF and LV dilation present in mtCaMKII hearts, as you noted. However, the interbred hearts remain significantly dilated (i.e. no rescue of end diastolic volume, EDV, in the interbred mice compared to the mtCaMKII mice) and have significantly reduced EF compared to AC3-I hearts, which exhibit slightly greater EF and slightly lower LV size compared to WT controls (Fig 2C). We conclude that the prominent, residual LV dilation and dysfunction in the mtCaMKII x AC3-I hearts represents the net effect of mitochondrial CaMKII over-expression.

6. The 2nd line of evidence is shown in Figure D but the authors do not show the percent of mtCaMKIIN that is in the cytosol vs. the mitochondria.

Thank you for raising this issue. We previously showed that mtCaMKIIN is nearly exclusively confined to the mitochondria in mtCaMKIIN transgenic hearts (Joiner, et al., Supplemental Figure 1b) (Joiner et al. 2012). Since these data were obtained many years before we performed the experiments in this study, we considered the possibility that this phenotype was not static, and so repeated the western blots to determine the quantity of mtCaMKIIN in the cytosol and mitochondria. We were surprised to find a significant portion of mtCaMKIIN residing in the cytosol (revised Fig S1B). This finding is noted in the revised manuscript (page 4, paragraph 1). Although we do not know why the distribution of mtCaMKIIN drifted in these mice over time (nearly a decade), we believe that the subcellular targeting of CaMKIIN in the

mtCaMKIIN transgenic mice is adequate for the points made in this manuscript: that the reduction in left ventricular dilation after MI is reduced, at least in part, by mitochondrial CaMKII inhibition, and that interbreeding of mtCaMKII mice with mtCaMKIIN but not AC3-I mice rescues left ventricular dilation in mtCaMKII hearts.

7. The data showing a difference in ATP and pCr levels is at a time with massive cardiac pathology. It's difficult to attribute this as a direct result of mtCaMKII and not secondary to the severe phenotype.

Thank you for this comment. We agree that the severe cardiomyopathy could contribute to the defective energetics in the mtCaMKII hearts. However, the inciting trigger is almost certainly related to mitochondrial CaMKII over-expression, given the nature of the mtCaMKII model. Unfortunately, the MRI studies are not feasible in the very young mice, at a time prior to myocardial dilation and dysfunction. In order to acknowledge this point, we revised the Discussion section (page 11, paragraph 1).

8. The data regarding diastolic calcium correction with ATP is nice (Fig 3E,F). Can the authors provide the same data with different pacing for amplitude, rise-time and decay? It's a little difficult to understand that if there is insufficient ATP to manage diastolic calcium (likely an effect at SERCA, since it is the most ATP dependent) how the individual transient decay is not affected.

Thank you for this suggestion. Based on your comments we performed additional analyses. These new calculations for peak Ca^{2+} amplitude and Ca^{2+} decay are provided in revised Fig S4B-C (page 7, paragraph 3). The Ca^{2+} decay rate is significantly slower in the mtCaMKII myocytes, but is restored by ATP supplementation. We interpret these new data to be consistent with SERCA impairment due to insufficient ATP in mtCaMKII cardiomyocytes.

9. Fig 4A-D. How well do the CKmito and CK-M antibodies differentiate these two isoforms? The blots shown here are from total heart lysates, so it is difficult to assess if CK in the mitochondria specifically is truly altered in mtCaMKII hearts, or whether the CKmito and CK-M antibodies might be picking up both myofibrillar and mitochondrial isoforms. Showing cytosolic vs. mitochondrial fractions would be helpful here or since there are not significant differences these data could be disregarded.

Thank you for this point. We tested both antibodies on mitochondrial and cytoplasmic fractions from mouse heart tissues. We found that the CKmito protein tracked with the mitochondrial marker (VDAC1) and CK-M tracked with the cytoplasmic marker (GAPDH) (Fig S5A; page 8, paragraph 2). We interpret these new data to indicate that the 2 antibodies are specific, in that they exclusively or primarily recognize their target proteins and do not cross react with the other CK isoform.

10. Is there a significant increase in CK protein in the CK-mito Tg mouse? Did the authors also examine mRNA? Is there a tag on the transgene that can be exploited?

There is an increase in CK protein in the CKmito transgenic mouse. In order to make this point more clearly, we added new data to the western blots in revised Fig 4E and F. There is not a tag on the transgenic CK, and we did not examine mRNA because the protein expression is increased, consistent with earlier reports using a similar model (Gupta et al. 2012). Responsive to your point, our new data confirm a modest increase in CKmito protein in the CKmito transgenic hearts. We are intrigued that an increase in CKmito was sufficient to substantially rescue mtCaMKII cardiomyopathy, and wonder if this finding could provide insights into future therapies.

11. Fig. 4. Is increased CKmito phosphorylation associated with any alterations in CKmito activity? And is loss or inhibition of CKmito alone sufficient to drive diminished mitochondrial energetics and contractile dysfunction/dilation seen in mtCaMKII mice? Is CKmito a direct phosphorylation target of mtCaMKII?

These are interesting questions. We are not aware of any studies that show phosphorylation of CKmito is associated with alterations in its activity. CKmito knockout mice on the same background as we used in the current study (C57Bl/6J) do not develop LV dysfunction, but have a compensatory increase in myofibrillar CK and mitochondrial volume (Lygate et al. 2009). Our view, supported by our phosphoproteomic studies (Table S1), is that there are myriad CaMKII target proteins in mitochondria. We interpret the rescue of mtCaMKII by interbreeding with CKmito mice to show that (modest) supplementation of CKmito is sufficient for therapeutic benefit. However, we do not interpret our data to suggest that CKmito loss would be sufficient to phenocopy mtCaMKII cardiomyopathy. CKmito was identified in our phosphoproteomics study as being significantly more phosphorylated in the mtCaMKII hearts compared to WT, but we do not know if the site identified is a direct target of CaMKII.

12. Fig. 4G. How does CKmito expression in mtCaMKII hearts recover both ATP content and PCr content, rather than just shifting the balance between the two? How does CKmito increase the total pool of high energy phosphate species in this context?

This is an interesting point. We did not initially anticipate that both ATP and PCr content would be increased by CKmito overexpression in mtCaMKII hearts. The observation that overexpression of CKmito (Fig 4G), but not CK-M (Fig S5C), rescues mtCaMKII mice and restores energetics suggests to us that it is not simply the CK enzyme in its role to balance PCr and ATP and set ATP/ADP and $\Delta G_{\sim\text{ATP}}$, but in addition perhaps its mitochondrial location, different structure than CK-M, or other factors that distinguish CKmito and allow it to rescue mtCaMKII hearts. We are not aware of studies demonstrating that CKmito impacts purine salvage pathways. Although beyond the scope of this report on mtCaMKII, we are pursuing studies to identify effects of CKmito in failing hearts and other energy deficient states beyond its balancing of PCr and ATP.

13. *Fig 5. Does CKmito expression in the mtCaMKII mice rescue Complex I activity as measured in-gel or by OCR, in addition to rescuing complex I expression? What is the link with complex I expression? How does the CKmito rescue have such a big effect on complex I formation?*

Thank you for this question. We performed new complex I activity measurements in isolated mitochondria, and found that the activity is not restored in mtCaMKII x CKmito compared to mtCaMKII (Fig 5I). At this point, we do not understand the link between expression of complex I components and activity, nor how CKmito may affect complex I expression or assembly. However, our new activity data (Fig 5I) suggest that complex I formation is not restored in the mtCaMKII hearts when CKmito is overexpressed even though one component of the complex (NDUFB8) has increased expression in the crossed hearts. We have clarified this point in the text (page 9, paragraph 2).

14. *How does the PDH activity increase in mtCaMKII hearts correlate with the increase in phosphor-PDH reported in the mass-spec data, since phosphor-PDH is inhibited? Is this the inhibitory site? Also, there was no change in mitochondrial calcium which is a strong physiological regulator of PDH activity.*

This is a good question. The phosphorylation site on PDH identified in the MS experiments (S331) is different from the well-known sites (S293, S300, S232) that regulate its activity via mitochondrial Ca^{2+} . We do not yet know the functional consequences of this new site on PDH activity; however, this determination will require extensive work that is beyond the scope of our study. We revised the Discussion (page 11, paragraph 2), to better highlight the uncertain functionality of these new phosphorylation sites on PDH activity.

Reviewer #2 (Remarks to the Author):

Luczak and colleagues found that mitochondrial CaMKII is activated upon myocardial infarction and that targeted inhibition of CaMKII in mitochondria can preserve cardiac function. They then generated a model of CaMKII overexpression in mitochondria and show a phenotype of ATP depletion and dilated cardiomyopathy which can be rescued by creatine kinase re-expression. This is an interesting manuscript describing effects of CaMKII on mitochondrial energy production in association with dilatation of the myocardium. Many findings are novel and advance the field.

Thank you for your positive comments and thorough review of our manuscript.

Major comments that should be addressed:

1. *Fig. 1A: To assess CaMKII activity in MI, additional measures other than pCaMKII should be used because pCaMKII as measure has been reported to be limited due to cross reactivity of most of the antibodies used in the field. Can the authors measure mitochondrial CaMKII phosphorylation targets as they do in Fig. 2G for mtCaMKII, e.g. MCU phosphorylation?*

Thank you for this point. Based on your suggestion, we performed new studies to measure MCU phosphorylation one week after MI surgery (revised Fig 1A-B; page 3, paragraph 2). These new data show that MCU phosphorylation is increased in hearts one week after MI compared to sham operated hearts. These data complement the original data on pCaMKII, and so provide further evidence of increased CaMKII activity in mitochondria after MI.

2. Moreover, looking more closely to the delta C / delta B ratio in Fig. 1A, it looks like a previous finding of Weinreuter et al in 2014 (EMBO Mol Med, Fig. 2A) that the ratio shifts towards delta C. Can the authors quantify this in their own experiment again?

This is an interesting point. We repeated the MI study and western blots in Fig 1A. The band corresponding to CaMKII δ does not clearly separate into two bands as seen in the previous study mentioned. Therefore, we do not feel comfortable making any claims regarding the $\delta C/\delta B$ ratio.

3. Can the authors show that hyperphosphorylation of the MCU upon MI is inhibited by mtCaMKIIN to judge whether CaMKII is indeed inhibited in this model?

Thank you for this suggestion. We performed additional MI surgeries and blotted for phosphorylated MCU (pMCU) 1 week after coronary ligation. We found a trend for reduced that pMCU in the mtCaMKIIN hearts compared to WT hearts (Figure S1D; page 4, paragraph 1). These data, taken together with the functional rescue of mtCaMKII mice by interbreeding with mtCaMKIIN mice, suggest that mitochondrial CaMKII is inhibited in the mtCaMKIIN transgenic heart, at baseline and after MI surgery.

4. Fig. 1C: The authors show that mtCaMKIIN mice are protected from cardiac dysfunction one week after myocardial infarction. But isn't that expected in light of the prior paper that these mice have less infarct size in the I/R model? Have the authors measured initial infarct size or troponin T levels in this experiment? This is important to assess the relative contribution of different CaMKII inhibition-dependent protective effects on cardiac function.

This is an important point. While it is true that mtCaMKIIN hearts are protected in an ex vivo I/R model, and acutely (5 hours) after MI, here we performed a permanent, chronic ligation MI, in vivo, and so did not anticipate a difference in MI size. However, based on your comments we repeated a series of MI studies (n = 9-10/group) using the same highly experienced surgeon who performed the original studies. In all studies she was blinded to the genetic identity of the mice. In these new studies, we measured infarct size in mtCaMKIIN and WT littermate hearts 24 hours after coronary artery ligation to assess the initial size of the injury (Fig S1C; page 4, paragraph 1). We saw no significant difference in the amount of infarcted myocardium. Additionally, we obtained echocardiographic measurements from a new cohort of mice (n=6-9/group) 1 week following coronary artery ligation and combined them with our original data set (revised Fig 1C). These new studies confirmed that the mtCaMKIIN expressing hearts were highly protected from myocardial dysfunction compared to WT littermate controls. In all cases the echocardiographer was blinded to the genetic identities of the mice. Taken together, our

data support a view that mtCaMKIIN protects against MI based on beneficial actions in surviving myocardium.

5. The authors refer to refs 15 and 21 to say that CaMKII is dispensable for cardiac function but in these studies only CaMKII delta was deleted and CaMKII activity was not completely abolished. They then correctly also refer to Kreuzer et al. (ref 76) because in this paper no residual CaMKII activity was detectable because the two redundant isoforms CaMKII delta and gamma were knocked out simultaneously. The same mouse model without any CaMKII activity was also used by Weinreuter et al. in an extensive ischemia/reperfusion study. How do the authors discuss their own findings in light of the Weinreuter paper? In that paper cardiac function upon I/R was protected only at later stages most likely to a reduced pro-inflammatory burden. Can the authors measure also pro-inflammatory cytokines and leucocyte infiltration in their model as alternative causes of contractile dysfunction?

Thank you for these comments. We agree that inflammation is a potentially important factor in myocardial dysfunction due to CaMKII activity, in myocardium and in other cell types. In order to determine if inflammation plays a role in our MI model, we repeated a series of MI surgeries and looked at inflammatory markers 1 week after coronary artery ligation. We found no significant differences in increased CCL2 or CCL3 mRNA (Fig S1E) or CD45+ cells (Fig S1F) in the infarcted area between mtCaMKIIN hearts and WT littermates. These data suggest that mitochondrial CaMKII activity does not significantly contribute to the inflammatory response in this model. These findings are noted in the revised manuscript (page 4, paragraph 2).

6. Loss of creatine as shown by the Neubauer lab does not cause exacerbation due to MI. How do the authors explain this in relation to the own findings?

This study by the Neubauer lab (Lygate et al. 2013) shows that loss of creatine due to knockout of the GAMT enzyme does not cause exacerbation of heart failure after MI. The authors also show that the hearts from the GAMT knockout mice have impaired contractile reserve and mitochondrial uncoupling.

In our current study, we show that mitochondrial creatine kinase (CKmito) is reduced in the hearts of the mtCaMKII mice, and that restoring CKmito expression rescues the impaired energetics, adverse remodeling, and dysfunction in mtCaMKII hearts. However, our view, supported by our phosphoproteomic studies, is that there are myriad CaMKII target proteins in mitochondria. We interpret the rescue of mtCaMKII by interbreeding with CKmito mice to show that (modest) supplementation of CKmito is sufficient for therapeutic benefit. However, we do not interpret our data to suggest that CKmito loss would be sufficient to phenocopy mtCaMKII cardiomyopathy. CKmito was identified in our phosphoproteomics study as being significantly more phosphorylated in the mtCaMKII hearts compared to WT, but we do not know if the site identified is a direct target of CaMKII.

Minor:

1. MW markers should be shown in the Western blot experiments

We have added molecular weight markers to all blot images.

In summary, I recommend a major revision.

Reviewer #3 (Remarks to the Author):

The paper by Luczak et al. investigates the role of mitochondrial calcium/calmodulin-dependent protein kinase II (CaMKII) in cardiac myocytes and its role in regulating mitochondrial function during pathology. First, the authors observed that one week after myocardial infarction (MI), mitochondrial CaMKII expression and phosphorylation (activity) is increased. Genetic inhibition of mitochondrial CaMKII protects from left ventricular (LV) dilation and dysfunction after MI. Mice with overexpression of mitochondrial CaMKII in the heart have dilated cardiomyopathy with energetic deficit. This can be rescued when mitochondrial creatine kinase (mitCK) is overexpressed in parallel. The authors conclude that these insights provide two new strategies how to prevent LV dilation after MI, i.e., increasing mitoCK or inhibiting mitochondrial CaMKII.

The study is generally very well conducted and addresses a clinically import topic, i.e., heart failure after MI, one of the most frequent causes for cardiovascular and total deaths worldwide. Furthermore, gaining insights into mitochondrial dysfunction after ischemia/reperfusion may also be of broader interest since similar mechanisms may (or may not) be in place after stroke or ischemia of other regions of the body.

The findings of the paper are indeed novel. The manuscript is nicely written, and the Figures are clear and well organized. A particular strength of the study is the combination of extensive genetic work on in vivo mouse models with functional experiments on isolated mitochondria and cardiac myocytes. The findings would have an important impact on the field, since the control of mitochondrial function in particular, after MI and in the context of maladaptive remodeling, is not fully understood, and CaMKII is a key regulatory hub that receives widespread attention in the scientific community.

However, the study has also several shortcomings that limit my enthusiasm about the conclusions drawn. These are i) the mere use of transgenic techniques to overexpress and target certain proteins in cardiac myocytes (which may be more artificial) rather than using knock-out technology, ii) the fact that no clear (molecular) mechanism of how CaMKII enters mitochondria upon stress and influences mitochondrial function under physiological or pathological situations is proposed, and iii) some of the conclusions drawn are not (yet) fully supported by the experiments. In the following, I will address these issues in more detail:

Thank you for your positive comments, constructive criticism, and thorough review of our manuscript. Our comments to your major and minor points follow.

Major:

1. Figure 1: The localization of CaMKII in mitochondria is not very convincing yet. It is a technical

challenge to isolate mitochondria without contamination by the ER/SR, based on a close interaction and tethering of these two organelles. Can you verify that the CaMKII expression and phosphorylation assessed in isolated mitochondria stems from a pure mitochondrial isolation? You show the presence of VDAC, verifying mitochondrial protein, but do not show the absence of SR-related proteins.

This is a good point. We have repeated the blots in Fig 1A with a new series of MI surgeries. We have included a marker of mitochondrial matrix (PDH), SR (SERCA2a), and cytoplasm (GAPDH) (revised Fig 1A). While SERCA2a is not detected in these new mitochondrial fraction blots, it was detected in the cytoplasmic fraction (Fig S1A). Additionally, we measured phosphorylated MCU (p-MCU), a mitochondrial CaMKII target (Nguyen et al. 2018), and show a significant increase in p-MCU 1 week after MI compared to sham hearts (revised Fig 1A-B; page 3, paragraph 2). These data further support that increased mitochondrial CaMKII persists one week after MI.

2. Another important information, also in view of your results on regulation of mitoCK and Krebs cycle enzymes by CaMKII (see below), is the actual localization of CaMKII within mitochondria (inner or outer mitochondrial membrane, matrix). In the introduction, you mention matrix localization. For this, further biochemical experiments with differential digestion protocols would be required to determine the actual localization of native CaMKII. The same applies to the transgenic overexpressed and targeted mitoCaMKII (Figure 2A). The immune-fluorescence colocalization of CaMKII with Cox IV (Figure 1E) alone hardly answers this, especially since CoxIV is in the inner mitochondrial membrane. If CaMKII resides indeed in the matrix, how would it be regulated from upstream, and would that be via signals stemming from inside or outside the mitochondria?

We apologize for the lack of clarity in our original manuscript. We have shown localization of endogenous mitochondrial CaMKII and mtCaMKIIN using mitochondrial sub-fractionation protocols and shown them to be present in the mitoplast fraction (Joiner et al. 2012). We have clarified this in the text (page 2, paragraph 2). Our transgenic models, mtCaMKIIN (mitochondrial CaMKII inhibition) and mtCaMKII (mitochondrial CaMKII over-expression) both use the highly validated Cox 8a TOM/TIM targeting sequence for localizing to the mitochondrial matrix. In addition to subcellular fractionation studies employed in our original studies, here we used novel approaches to test the fidelity of mitochondrial targeting of the over-expressed mtCaMKII, and its role in driving the dilated cardiomyopathy phenotype. First we crossed the mtCaMKII mice with established mice with transgenic expression of a CaMKII inhibitory peptide (AC3-I). For unknown reasons, the transgenically expressed AC3-I/eGFP fusion protein is excluded from mitochondria (see Fig 2B), and only resides in the cytoplasm and nucleus (Zhang et al. 2005). We exploited this pattern of subcellular distribution by crossing AC3-I with mtCaMKII mice; these interbred mice with mitochondrial CaMKII over-expression and cytoplasmic and nuclear CaMKII inhibition due to AC3-I expression exhibited a dilated cardiomyopathy (see Fig 2C), similar to mtCaMKII mice, indicating the mitochondrial-targeted

component of mtCaMKII was responsible for dilated cardiomyopathy. In contrast, the modest myocardial hypertrophy seen in mtCaMKII hearts was reversed by interbreeding with AC3-I (Fig 2C), suggesting that myocardial hypertrophy in mtCaMKII mice was a consequence of extramitochondrial expression 'leak' of CaMKII. Additionally, we crossed mtCaMKIIN and mtCaMKII mice; these interbred mice did not have dilated cardiomyopathy (Fig 2D). Taken together, these results strongly support a view that the mitochondrial targeting of CaMKII in mtCaMKII mice is critical for driving dilated cardiomyopathy but not myocardial hypertrophy.

We interpret our studies, using orthogonal approaches, to show a key role for mitochondrial-targeted CaMKII in promoting dilated cardiomyopathy due to depressed energetics, independent of myocardial hypertrophy and death. Based on the biology of CaMKII, we assume that mitochondrial Ca^{2+} , ROS, and, perhaps, *O*-GlcNAc are candidate upstream signals for regulating mitochondrial CaMKII activity. These signals are all abundant in the mitochondrial matrix. We are pursuing studies to discover the mechanisms for CaMKII import to mitochondria, but these are not complete and are outside the scope of this report. We have edited the text to make these points more clear (page 12, paragraph 2).

3. The transgenic overexpression of CaMKII in mitochondria is rather an artificial condition. This should be at least discussed, since in this project, no KO strategy is used, which would provide a somewhat cleaner information. This is of particular relevance because in some aspects, your previous results on reduction of infarct size by mitochondrial CaMKII inhibition differ from observations in CaMKII delta/gamma double KO mice (Weinreuter et al., EMBO Mol Med 2014), in which infarct size was unaffected by the KO (although the subsequent remodeling was ameliorated).

These are excellent points, and we added a brief discussion on the potential limitations of transgenic mitochondrial CaMKII over-expression (page 14, paragraph 2). While we agree that a knock out strategy brings some advantages, we do not know how we could 'target' a knock out exclusively, or predominantly, to the mitochondrial matrix. Our previous study showed reduction of infarct size in mtCaMKIIN hearts (i.e. hearts with myocardial and mitochondrial matrix-targeted CaMKII inhibition) following I/R injury, and acute (5 hour) MI (Joiner et al. 2012). The I/R studies were performed ex vivo on a Langendorff apparatus, and infarct size was determined acutely. In Weinreuter et al., mice were subjected to in vivo I/R surgery and infarct size was determined 24 hours later, so these results were obtained under substantially different experimental conditions. In contrast to either the Joiner or Weinreuter papers, our new studies use a permanent coronary artery ligation model with a focus on times 1-3 weeks after the MI. Thus, we do not believe our current data directly address these, important, earlier studies.

MI surgery has the potential for variability. Based on your comments and related comments by Reviewer 1, we repeated a series of MI studies (n = 9-10/group) using the same highly experienced surgeon who performed the original studies. In all studies she was blinded to the genetic identity of the mice. In these new studies, we measured infarct size in mtCaMKIIN and WT littermate hearts 24 hours after coronary artery ligation to assess the initial size of the

injury (revised Fig S1C and revised Supplemental Methods; Supplemental Information page 2). We saw no significant difference in the amount of infarcted myocardium. Additionally, we obtained echocardiographic measurements from a new cohort of mice (n=6-9/group) 1 week following coronary artery ligation. These new studies confirmed that the mtCaMKIIN expressing hearts were highly protected from myocardial dysfunction compared to WT littermate controls. In all cases the echocardiographer was blinded to the genetic identities of the mice. These new data were combined with the previous cohort and shown in revised Fig 1C. Taken together, we interpret our data to suggest that mtCaMKIIN improves cardiac function after MI by improving the surviving myocardium rather than reducing the infarct size.

4. Figure 2I-J: The observation that overexpression of mitochondrial CaMKII does not affect MCU-mediated Ca uptake stands at odds with your previously proposed concept in which CaMKII actually does control MCU-mediated mitochondrial Ca uptake (Joiner, Nature). In this context, did you also re-test whether CaMKII-IN expression affects mitochondrial Ca uptake? And from a technical side, you may get a cleaner view on the actual mitochondrial Ca uptake if you perform the Ca retention assays on isolated mitochondria in the presence of cyclosporine A, since differential PTP activation (potentially controlled by CaMKII?) could blur your view on uptake by affecting Ca release via PTP. Therefore, I suggest to perform these experiments in the absence and presence of CsA.

We did initially anticipate mtCaMKII mitochondria would show increased Ca²⁺ uptake/content; for this reason we created a new model by interbreeding the mtCaMKII with cyclophilin knock out (*Ppif*^{-/-}) mice (Fig S3C). As we reported, and to our initial surprise, reducing mPTP opening responses to Ca²⁺ did not rescue the dilated cardiomyopathy phenotype in mtCaMKII mice. Instead, our studies indicated that chronic mitochondrial CaMKII over-expression resulted in diminished energetics that we ultimately were able to rescue by interbreeding with CKmito mice. However, we now consider that our initial hypothesis may have been naïve in that chronic over-expression of mitochondrial CaMKII is fundamentally different than chronic mitochondrial CaMKII inhibition, and could certainly lead to compensations that prevented our anticipated changes to mitochondrial Ca²⁺. Thus, we did not retest the mtCaMKIIN myocytes in the mitochondrial Ca²⁺ uptake assays, and do not think the results in mtCaMKII myocytes necessarily contradict our original findings. However, we did perform the requested mitochondrial Ca²⁺ uptake assays in the presence of CsA, and have included the data in revised Fig 2I-J (page 6, paragraph 2). The addition of CsA did induce increased mitochondrial Ca²⁺ retention in WT mitochondria, and a modest reduction in apparent rate of Ca²⁺ uptake in mtCaMKII compared to WT mitochondria. These new data, along with the genetic cross with the *Ppif*^{-/-} mice presented in Fig S3C, suggest that chronic mitochondrial CaMKII over-expression may have a mild protective affect against opening of the mPTP.

5. Figure 2L: In the methods you mention sequential additions of pyruvate/malate, rotenone and antimycin A. Which of these conditions do you show in the figure? Can you show all? Did you also measure H2O2 emission in the presence of ADP to simulate more physiological (respiring)

conditions? Mitochondrial H₂O₂ emission is controlled also by NADPH and the transhydrogenase (Nnt), which is not expressed in BL/6J mice. Do you know about the expression of the Nnt in your various mouse models? This may have an impact on the results of H₂O₂ emission and, if indeed most are in BL/6J background, should be at least critically discussed.

Thank you for these questions. In figure 2L we only show H₂O₂ emission after addition of pyruvate/malate. We have included the data to show H₂O₂ emission after addition of rotenone and antimycin A in revised Fig 2L. There is no significant difference between WT and mtCaMKII under any of these conditions. We did not measure H₂O₂ emission in the presence of ADP. While this would increase the rate of emission, we don't anticipate we would detect a difference in the mtCaMKII mitochondria.

We are aware that the C57Bl/6J strain does not express *Nnt*, and that this leads to reduced mitochondrial ROS production in pathological models (Nickel et al. 2015). We bred all our lines onto the C57Bl/6J background, and used littermate controls in all our experiments to ensure that any differences in expression due to gene mutations is minimal. In order to verify that the mtCaMKII phenotype is not strain-specific, we back-crossed the line 5 generations to the CD1 background. We found a similar reduction in cardiac function and increased left ventricular dilation in the mtCaMKII mice compared to littermate controls (revised Fig S3B) (page 6, paragraph 2). Additionally, mouse lines purchased from Jackson Labs (*mCat*, *Ppif*^{-/-}) were not available on the C57Bl/6J background. Littermate controls from these crosses also had a similar phenotype as the original C57Bl/6J line when expressing the mtCaMKII transgene (compare mtCaMKII in Fig S3A and C with mtCaMKII in Fig 1G). Given these data, we do not believe that *Nnt* expression is playing a major role in the development of the dilated heart phenotype in the mtCaMKII mice.

6. Figure 3: Why did you use Cs-based bath and pipette solutions in your experiments determining cytosolic Ca transients? Would it not be better to allow EC coupling as during physiological conditions? And since the in vivo phenotype is a systolic form of heart failure, I would be interested in the amplitudes and kinetics of cytosolic calcium transients as well; from the traces you show one could imagine that in the mitoCaMKII, they are somewhat reduced?

We used voltage clamp and Cs⁺ containing solutions, rather than field stimulation, to directly measure cytoplasmic Ca²⁺ transients, in order to measure changes independently of potential electrical remodeling in the cardiomyopathic mtCaMKII cardiomyocytes, and to dialyze AIP and ATP into the cytoplasm for various studies.

Based on your comments, we added new analyses of our data. These new data are presented in revised Fig S4B-C. We did find that the cytoplasmic Ca²⁺ transients resolved more slowly in mtCaMKII than WT ventricular myocytes, and that ATP dialysis hastened the cytoplasmic Ca²⁺ transient decay rate in mtCaMKII, but not in WT cardiomyocytes. We found that Ca²⁺ transient amplitudes were increased by ATP dialysis in mtCaMKII but not in WT cardiomyocytes (page 7, paragraph 3). We interpret the reduced diastolic Ca²⁺ (Fig 3F), increased Ca²⁺ transient

amplitudes (Fig S4B) and hastened decline in the Ca²⁺ transient (revised Fig S4C) after ATP in mtCaMKII, but not in WT cardiomyocytes, as supporting the hypothesis that depressed energetics in mtCaMKII hearts contributed to mechanical dysfunction, at least in part, by reduced SR Ca²⁺ uptake and depressed cytoplasmic Ca²⁺ transients.

7. You mostly argue with reduced ATP, but you may consider that also an accumulation of ADP per se has a substantial impact on diastolic function (see Sequeira et al., J Physiol 2015), since ADP can induce diastolic dysfunction on the sarcomere level. Also, a change in the ATP/ADP ratio and thereby, delta G ATP may impact the rate of SERCA, and SERCA function may be reflected by the rates of calcium decay in your cell experiments. Therefore, please add the kinetics to these results.

Thank you for making these points. Based on your comments we performed additional analysis on the single cell studies (also discussed in response to point 6, above). These new calculations for peak Ca²⁺ amplitude and Ca²⁺ decay are now provided in Fig S4B-C. The Ca²⁺ decay rate is significantly slower in the mtCaMKII myocytes, but is restored by ATP supplementation. We interpret these new data to be consistent with SERCA impairment due to insufficient ATP in mtCaMKII cardiomyocytes. However, our results do not exclude other contributory factors. Thus, we agree that ADP accumulation could potentially contribute to the cardiomyopathy in mtCaMKII mice, and have added a brief discussion to this effect in the revised Discussion (page 13, paragraph 1).

8. Figure 5: Do you have any idea how complex I could be downregulated so strongly by mtCaMKII overexpression, and how changes of ATP could upregulate the expression of complex I (since CKmito expression partially improves complex I expression)?

At this point, we do not understand the link between expression of mtCaMKII and complex I components and activity. However, we did note in our phosphoproteomic studies that some complex I proteins show increased phosphorylation in mtCaMKII compared to WT hearts, suggesting a possible link between mtCaMKII and complex I protein turnover, assembly, and/or activity (see Table S1).

In order to better address this question about the link between CKmito and complex I, we performed new complex I activity measurements in isolated mitochondria, and found that the activity is not restored in mtCaMKII x CKmito compared to mtCaMKII (Fig 5I). At this point, we do not understand the link between expression of complex I components and activity, nor how CKmito may affect complex I expression or assembly. The new activity data suggests that complex I formation is not restored in the mtCaMKII hearts when CKmito is overexpressed even though one component of the complex (NDUFB8) has increased expression in the crossed hearts. We have clarified this point in the text (page 9, paragraph 2).

9. Figure 6: Your results of NADH/NAD+ in resting cardiac myocytes make sense, since in light of the upregulation of TCA cycle enzymes and downregulation of complex I, one indeed expects a more reduced redox state of NADH/NAD+. However, I did not understand why NADH becomes

more oxidized in response to pacing, which is even more a situation of electrons flowing from the Krebs cycle into the respiratory chain and thereby, should aggravate any alterations in redox state which may be related to the alterations in complex I and Krebs cycle enzymes, respectively. Furthermore, one can assume that during pacing, the actual redox state of NADH/NAD⁺ is similar between both groups, since the WT start more oxidized but become more reduced, while the mtCaMKII start more reduced but become oxidized. So what is the net steady-state redox state of NADH/NAD during pacing? Finally, if you collect also the redox state of FADH₂/FAD in these myocytes, you may get a good idea how your complex II upregulation affects the redox state of that pool (since FADH₂ fuels into complex II).

The increase in NADH with pacing for WT is consistent with some (White et al. 1993, Jo et al. 2006) but not all (Griffiths et al. 1998) previous work. Variation in published results likely reflects differences in the preparations (e.g. mechanically loaded versus unloaded, isolated cells versus tissue) and intracellular conditions (e.g. Ca²⁺, ADP, O₂, etc). We agree with your interpretation of the increased basal NADH in mtCaMKII ventricular myocytes. Our hypothesis is that the NADH generating capacity of TCA cycle dehydrogenases is maximal in mtCaMKII hearts, consistent with our findings that these enzymes show hyperphosphorylation (Table S1) and increased activity (Fig 6B).

Based on your comments we reanalyzed our data to show absolute changes in NADH between resting and paced conditions (revised Fig 6D). We did not measure FADH. We found that mtCaMKII myocytes have significantly more NADH compared to WT myocytes at baseline, and that this difference goes away after pacing. Additionally, WT myocytes significantly increase NADH levels when paced, while mtCaMKII myocytes have significantly decreased NADH when paced.

10. Figure 7: The ATP/ADP ratio is rather small, I would expect values of >100?

We used published values of ATP/ADP_{total} measurement for our modelling (Tantama et al. 2013). These resulted in an ATP/ADP ~40. We recognize that most ADP is bound and that the metabolically relevant portion is the cytosolic free ADP (ADP_{free}), which is much lower (Zhou et al. 2006). Thus, a calculation based on free ADP would give an ATP/ADP_{free} that is higher, or approximately 88. We reran the simulation with this altered parameter. The findings remain the same showing reduced ATP/ADP_{free} in the mtCaMKII hearts compared to WT, and a return to near WT levels in the mtCaMKII x CKmito crossed hearts. We have updated the figure to reflect this adjustment (revised Figure 7B).

11. You mention that in mtCaMKII mice, CKmito is hyperphosphorylated. How specific is this result? Is there a known phosphorylation site of CK by CaMKII? Is this phosphorylation blunted in CaMKII KO mice? And is CKmito downregulated in HF (for instance, in your MI model)?

This is an interesting point. We believe the result is specific because we measured it in biological replicates, using mitochondria isolated from WT and mtCaMKII hearts. At this point, we do not know if this phosphorylation is present in other models, or if it contributes to

downregulation of CKmito or loss of CKmito activity. We quantified CKmito expression in hearts from mice 1 week after MI or sham surgery. We found no difference in CKmito in the infarcted hearts, suggesting that mtCaMKII over-expression is a stronger negative regulator of CKmito expression than MI, at least under these experimental conditions. We mention this new result in the revised manuscript (Fig S5B; page 8, paragraph 2).

References

Griffiths, E. J., H. Lin and M. S. Suleiman (1998). "NADH fluorescence in isolated guinea-pig and rat cardiomyocytes exposed to low or high stimulation rates and effect of metabolic inhibition with cyanide." Biochem Pharmacol **56**(2): 173-179.

Gupta, A., A. Akki, Y. Wang, M. K. Leppo, V. P. Chacko, D. B. Foster, V. Caceres, S. Shi, J. A. Kirk, J. Su, S. Lai, N. Paolocci, C. Steenbergen, G. Gerstenblith and R. G. Weiss (2012). "Creatine kinase-mediated improvement of function in failing mouse hearts provides causal evidence the failing heart is energy starved." J Clin Invest **122**(1): 291-302.

Jo, H., A. Noma and S. Matsuoka (2006). "Calcium-mediated coupling between mitochondrial substrate dehydrogenation and cardiac workload in single guinea-pig ventricular myocytes." J Mol Cell Cardiol **40**(3): 394-404.

Joiner, M. L., O. M. Koval, J. Li, B. J. He, C. Allamargot, Z. Gao, E. D. Luczak, D. D. Hall, B. D. Fink, B. Chen, J. Yang, S. A. Moore, T. D. Scholz, S. Strack, P. J. Mohler, W. I. Sivitz, L. S. Song and M. E. Anderson (2012). "CaMKII determines mitochondrial stress responses in heart." Nature **491**(7423): 269-273.

Lygate, C. A., D. Aksentijevic, D. Dawson, M. ten Hove, D. Phillips, J. P. de Bono, D. J. Medway, L. Sebag-Montefiore, I. Hunyor, K. M. Channon, K. Clarke, S. Zervou, H. Watkins, R. S. Balaban and S. Neubauer (2013). "Living without creatine: unchanged exercise capacity and response to chronic myocardial infarction in creatine-deficient mice." Circ Res **112**(6): 945-955.

Lygate, C. A., I. Hunyor, D. Medway, J. P. de Bono, D. Dawson, J. Wallis, L. Sebag-Montefiore and S. Neubauer (2009). "Cardiac phenotype of mitochondrial creatine kinase knockout mice is modified on a pure C57BL/6 genetic background." Journal of Molecular and Cellular Cardiology **46**(1): 93-99.

Nguyen, E. K., O. M. Koval, P. Noble, K. Broadhurst, C. Allamargot, M. Wu, S. Strack, W. H. Thiel and I. M. Grumbach (2018). "CaMKII (Ca²⁺)/Calmodulin-Dependent Kinase II) in Mitochondria of Smooth Muscle Cells Controls Mitochondrial Mobility, Migration, and Neointima Formation." Arterioscler Thromb Vasc Biol.

Nickel, Alexander G., A. von Hardenberg, M. Hohl, Joachim R. Löffler, M. Kohlhaas, J. Becker, J.-C. Reil, A. Kazakov, J. Bonnekoh, M. Stadelmaier, S.-L. Puhl, M. Wagner, I. Bogeski, S. Cortassa, R. Kappl, B. Pasięka, M. Lafontaine, C. Roy D. Lancaster, Thomas S. Blacker, Andrew R. Hall, Michael R. Duchon, L. Kästner, P. Lipp, T. Zeller, C. Müller, A. Knopp, U. Laufs, M. Böhm, M. Hoth and C. Maack (2015). "Reversal of Mitochondrial Transhydrogenase Causes Oxidative Stress in Heart Failure." Cell Metabolism **22**(3): 472-484.

Tantama, M., J. R. Martinez-Francois, R. Mongeon and G. Yellen (2013). "Imaging energy status in live cells with a fluorescent biosensor of the intracellular ATP-to-ADP ratio." Nat Commun **4**: 2550.

White, R. L. and B. A. Wittenberg (1993). "NADH fluorescence of isolated ventricular myocytes: effects of pacing, myoglobin, and oxygen supply." Biophys J **65**(1): 196-204.

Zhang, R., M. S. Khoo, Y. Wu, Y. Yang, C. E. Grueter, G. Ni, E. E. Price, Jr., W. Thiel, S. Guatimosim, L. S. Song, E. C. Madu, A. N. Shah, T. A. Vishnivetskaya, J. B. Atkinson, V. V. Gurevich, G. Salama, W. J. Lederer, R. J. Colbran and M. E. Anderson (2005). "Calmodulin kinase II inhibition protects against structural heart disease." Nat Med **11**(4): 409-417.

Zhou, L., M. E. Cabrera, I. C. Okere, N. Sharma and W. C. Stanley (2006). "Regulation of myocardial substrate metabolism during increased energy expenditure: insights from computational studies." Am J Physiol Heart Circ Physiol **291**(3): H1036-1046.

REVIEWER COMMENTS

Reviewer #1 (Remarks to the Author):

Thank you for a thoughtful response to critique. The new data has greatly improved the manuscript.

Reviewer #2 (Remarks to the Author):

This is an extensive revision. All points have been addressed and the manuscript has improved. This is an important study that further contributes to the understanding of CaMKII in cardiac pathophysiology. I have no further questions.

Johannes Backs

Reviewer #3 (Remarks to the Author):

The authors have extensively revised their manuscript, addressed all points raised by this and the other reviewers and have added new experiments to support their data and conclusions.

I have no further questions.

Christoph Maack

Reviewer #4 (Remarks to the Author):

The Authors present a three-compartment mitochondrial model, based on previous work, that incorporates major processes in addition to the Creatine-Phosphocreatine system that regenerates ADP, both in cytoplasmic and periplasmic spaces.

The Authors simulate the metabolic phenotype of the mitochondrial CaMKII (mtCaMKII) and mitochondrial CK overexpressors by, e.g., decreasing complex I activity or mitochondrial CK, increasing the activity of TCA cycle enzymes (Fig. 7).

Two important remarks that the Authors should address:

1) They simulate the cycle of ATP hydrolysis occurring during the heart cycle of systole-diastole (their Fig. S6). However, they do not do the same with the cytoplasmic Ca^{2+} (Cai), and they should, otherwise it won't be a realistic simulation since Ca^{2+} also oscillates during the cycle of contraction-relaxation of the heart. Apparently, the Authors left Cai constant, as a parameter, after inspection of the model equations (in Gitlab) by this Reviewer. As a result, the mitochondrial Ca^{2+} (Camito) appears not to change (or very little) which has metabolic consequences for the flux through the TCA cycle, respiration and ATP synthesis, all relevant for the present work.

2) Although the simulation approach utilized is correct, its reliability depends on the model parameterization under conditions as close as possible to the experimental situation being simulated, to validate the model behavior under those conditions. This should be accompanied by a Supplementary Table with the parameters utilized in the simulations and their references for justification.

Do the Authors have simulations of relevant experimental results from this work or the work of others, independently from what they are trying to simulate, e.g., ATP/ADP or ATP/PCr ratio? If so, please, include in Supplement.

We would like to thank you for your summary, comments, and thorough review of our manuscript and model. Here we address the critiques in a point-by-point fashion. Text responses and changes to the original manuscript are marked by yellow highlighting.

Reviewer #4 (Remarks to the Author):

The Authors present a three-compartment mitochondrial model, based on previous work, that incorporates major processes in addition to the Creatine-Phosphocreatine system that regenerates ADP, both in cytoplasmic and periplasmic spaces.

The Authors simulate the metabolic phenotype of the mitochondrial CaMKII (mtCaMKII) and mitochondrial CK overexpressors by, e.g., decreasing complex I activity or mitochondrial CK, increasing the activity of TCA cycle enzymes (Fig. 7).

Two important remarks that the Authors should address:

1) They simulate the cycle of ATP hydrolysis occurring during the heart cycle of systole-diastole (their Fig. S6). However, they do not do the same with the cytoplasmic Ca²⁺ (Cai), and they should, otherwise it won't be a realistic simulation since Ca²⁺ also oscillates during the cycle of contraction-relaxation of the heart. Apparently, the Authors left Cai constant, as a parameter, after inspection of the model equations (in Gitlab) by this Reviewer. As a result, the mitochondrial Ca²⁺ (Camito) appears not to change (or very little) which has metabolic consequences for the flux through the TCA cycle, respiration and ATP synthesis, all relevant for the present work.

Thank you for raising this important issue regarding calcium cycling. We agree that calcium is an important energetic regulator in the heart, and that mitochondrial calcium plays a role in regulating ATP production through activating TCA cycle enzymes. We have rerun the simulations adding cytosolic calcium pulses to simulate the cardiac calcium cycles. We used the data generated for Figures 3E and 4H to model the calcium pulses (see Figures 7 and S6, and Table S2). A small fluctuation of mitochondrial calcium occurs in the modified simulations, and is in agreement with another published model (Song et al. 2019), and published experimental data (Lu et al. 2013). The other parameters measured did not significantly change as a result of including the cytosolic calcium pulses.

2) Although the simulation approach utilized is correct, its reliability depends on the model parameterization under conditions as close as possible to the experimental situation being simulated, to validate the model behavior under those conditions. This should be accompanied by a Supplementary Table with the parameters utilized in the simulations and their references for justification.

Do the Authors have simulations of relevant experimental results from this work or the work of others, independently from what they are trying to simulate, e.g., ATP/ADP or ATP/PCr ratio? If so, please, include in Supplement.

Thank you for your comments. We have now provided an additional document for the computational simulations, including detailed model formulations, parameters, simulation conditions, and initial conditions (see Supplemental Information, Model Validation). The model is validated from the experimental results from this paper (Table S2) as well as from the literature (see Supplemental Information, Model Validation). Generally speaking, parameters were taken from the literature and were supplemented with our own experimental data to simulate the conditions in the different genotypes.

References

Lu, X., K. S. Ginsburg, S. Kettlewell, J. Bossuyt, G. L. Smith and D. M. Bers (2013). "Measuring Local Gradients of Intramitochondrial $[Ca^{2+}]$ in Cardiac Myocytes During Sarcoplasmic Reticulum Ca^{2+} Release." Circulation Research **112**(3): 424-431.

Song, Z., L.-H. Xie, J. N. Weiss and Z. Qu (2019). "A Spatiotemporal Ventricular Myocyte Model Incorporating Mitochondrial Calcium Cycling." Biophysical Journal **117**(12): 2349-2360.

REVIEWERS' COMMENTS:

Reviewer #4 (Remarks to the Author):

The Authors have satisfactorily addressed my comments.